# COMPUTING EX ANTE EQUILIBRIUM IN HETEROGENEOUS ZERO-SUM TEAM GAMES

## ABSTRACT

The *ex ante* equilibrium for two-team zero-sum games, where agents within each team collaborate to compete against the opposing team, is known to be the best a team can do for coordination. Many existing works on *ex ante* equilibrium solutions are aiming to extend the scope of *ex ante* equilibrium solving to large-scale team games based on Policy Space Response Oracle (PSRO). However, the joint team policy space constructed by the most prominent method, Team PSRO, cannot cover the entire team policy space in *heterogeneous* team games where teammates play distinct roles. Such insufficient policy expressiveness causes Team PSRO to be trapped into a sub-optimal *ex ante* equilibrium with significantly higher exploitability and never converges to the global *ex ante* equilibrium. To find the global *ex ante* equilibrium without introducing additional computational complexity, we first parameterize heterogeneous policies for teammates, and we prove that optimizing the heterogeneous teammates' policies sequentially can guarantee a monotonic improvement in team rewards. We further propose **Heterogeneous-PSRO** (**H-PSRO**), a novel framework for *heterogeneous* team games, which integrates the sequential correlation mechanism into the PSRO framework and serves as the first PSRO framework for *heterogeneous* team games. We prove that H-PSRO achieves lower exploitability than Team PSRO in *heterogeneous* team games. Empirically, H-PSRO achieves convergence in matrix heterogeneous games that are unsolvable by non-heterogeneous baselines. Further experiments reveal that H-PSRO outperforms non-heterogeneous baselines in both heterogeneous team games and homogeneous settings.

## 1 INTRODUCTION

In this paper, we focus on a class of multiplayer games where a team of agents competes against an adversarial team. Specifically, we focus on a team of heterogeneously skilled agents against an opposing team, which is referred to as *heterogeneous* team games. These games model competitions between two entities (the team and the opponent team) and are natural extensions of two-player games to multiplayer games. To this day, algorithms have achieved superhuman performance in two-player games, including Go (Silver et al., 2017) and heads-up no-limit Texas hold'em poker games (Bowling et al., 2015; Brown & Sandholm, 2018).

*Heterogeneous* team games introduce distinct challenges not presented in two-player games, especially in terms of coordinating team members with distinct roles. For example, in StarCraft, how should distinct species (e.g., Marine, Stalker, and Medivac), each with unique skills, collaborate to defeat an opposing team? Similarly, in soccer, how can forwards, midfielders, and defenders, who cannot communicate during the game except through public observable actions, play optimally against their opponents? A common solution for these challenges is that of *ex ante* coordination introduced by Celli & Gatti (2018), in which team members can correlate their strategies before the game starts but cannot communicate during the gameplay. This form of *ex ante* coordination is known to be the best a *heterogeneous* team can do for coordination and makes the problem of optimization convex. However, computing an *ex ante* equilibrium in *heterogeneous* team games is inapproximate in polynomial time (Celli & Gatti, 2018) even though it is equivalent to a Nash equilibrium in a two-player zero-sum game (McAleer et al., 2023).

Existing methods for two-team zero-sum games either can solve only small to mediated-sized *heterogeneous* team games, or can scale to large games but only in a setting of homogeneous team-

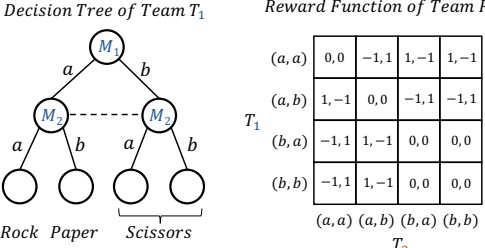

| | Decision Tree of Team $T_1$ | Reward Function of Team RPS | Homogeneous PSRO framework in Team RPS |

**Pure strategy (a, b) of both $T_1$ and $T_2$** cannot be represented by the **homogeneous PSRO framework**

Figure 1: Procedure of the homogeneous PSRO framework in Team Rock-Paper-Scissors, which is a typical *heterogeneous* team game, with four agents, two teams $T_1 = \{M_1, M_2\}$ and $T_2 = \{O_1, O_2\}$, one state, and joint action spaces $\mathcal{A}_1 = \mathcal{A}_2 = \{a, b\}^2$. Agents play Rock-Paper-Scissors between the teams: if player $M_1$ in team $T_1$ (or $O_1$ in team $T_2$) chooses action $b$, then the team plays $Scissors$ no matter the choice of the other player in the team; if both players choose action $a$, then the team plays $Rock$; otherwise, the team plays $Paper$. The two players in team $T_1$ or opponent team $T_2$ are *heterogeneous* because the actions $a$ and $b$ serve different functions for them. Specifically, player $M_1$ (or $O_1$) can unilaterally choose the team decision $Scissors$ by playing action $b$, while player $M_2$ (or $O_2$) must coordinate with the other player to choose $Paper$ by playing action $b$.

mates. On one hand, some researchers model the problem of solving *ex ante* equilibria in two-team zero-sum games as a linear program and resort to Column Generation (CG) algorithms (Celli & Gatti, 2018; Farina et al., 2018; Zhang & An, 2020; Farina et al., 2021; Zhang et al., 2021; Zhang & Sandholm, 2022; Zhang et al., 2022a; Zhang et al., 2022b; Zhang et al., 2024) to solve it. CG algorithms iteratively compute a joint distribution as the optimal *ex ante* team coordination strategy and can be naturally extended to solve *heterogeneous* team games. However, since the jointly coordinated team strategy space grows exponentially with the increasing number of teammates, scaling CG to large heterogeneous team games becomes challenging. On the other hand, McAleer et al. (2023) solve large-scale team games by extending a reinforcement learning-based equilibrium solver–Policy Space Response Oracle (PSRO) (Lanctot et al., 2017)–from two player zero-sum games to two-team zero-sum games and utilizing a homogeneous agent-based method for team coordination. This homogeneous method computes the optimal *ex ante* team coordination strategy by sharing a policy among teammates, and enables efficient team coordination as the increasing number of agents does not introduce much computational and sample complexity burden. However, this homogeneous PSRO framework (McAleer et al., 2023) cannot converge in *heterogeneous* team games.

We identify and analyze the convergence issues of early termination and sub-optimal equilibrium trap that homogeneous PSRO framework encounters in *heterogeneous* team games, and show that the primary reason is its heavy reliance on the policy sharing mechanism for team coordination. Specifically, the policy sharing mechanism requires players in both teams to be homogeneous (e.g., share the same observation space and action space, and play similar roles in a cooperation task). When applied to *heterogeneous* team games, this mechanism cannot represent all the team policies. For example, a pure strategy of joint action $(a, b)$ in the team joint action space $\{a, b\} \times \{a, b\}$ is excluded from team policies with a shared distribution $(x, 1 - x)$, because it requires $x = 1$ and $1 - x = 1$ to hold at the same time. Since an *ex ante* equilibrium is a pair of correlated team strategies, such an **insufficient *policy expressive ability*** further makes the equilibrium space with policy sharing unable to cover the whole equilibrium space, leading to **insufficient *equilibrium expressive ability***. As a result, the homogeneous PSRO framework can only converge to a **sub-optimal equilibrium** with significantly higher exploitability in *heterogeneous* team games.

In *heterogeneous* team games, the homogeneous PSRO framework may terminate early, being trapped into a sub-optimal ex ante equilibrium with significantly higher exploitability, because the Best Response policy does not exist in the team policy space under the condition of insufficient *policy expressiveness*. For example, in a typical heterogeneous team game Team *Rock-Paper-Scissors* in Figure 1, the Best Response to a meta policy of the pure strategy Rock cannot be represented by the policy sharing mechanism, making the homogeneous framework trapped in a Rock policy and never find the global *ex ante* equilibrium. To address the above convergence issue caused by insufficient *policy expressiveness*, one straightforward idea is to parameterize heterogeneous policies for each player. That is, we jointly optimize over multiple players' policy spaces to find an optimal *ex ante* coordination strategy. However, optimizing over multiple players' policy spaces simultane-

ously is significantly harder than optimizing over a single shared player's policy space (Zhang et al., 2022b). To find the global *ex ante* equilibrium without introducing high computational complexity, we serialize the optimization process of heterogeneous policies and prove that optimizing the heterogeneous policies sequentially can guarantee a monotonic improvement on the team rewards. With this sequential correlation, the policy spaces of two teams grow linearly, avoiding the exponential increase with the increasing number of agents. Inspired by these results, we propose a novel framework for *heterogeneous* team games, Heterogeneous-PSRO (**H-PSRO**), which integrates the sequential correlation mechanism into an iterative procedure and serves as the first PSRO framework for computing an *ex ante* equilibrium in *heterogeneous* team games. We prove that H-PSRO achieves lower exploitability than Team PSRO in *heterogeneous* team games. As a result, H-PSRO framework shows empirical convergence in matrix heterogeneous games that are unsolvable by its homogeneous-counterparts. Further results on a suite of large scale benchmark games show that H-PSRO can be implemented for large scale games and surprisingly outperforms the homogeneous framework in not only heterogeneous team games, but also homogeneous settings.

## 2 PRELIMINARIES

Two-team zero-sum game (Littman, 1994) [1] can be defined as a tuple $G = (\mathcal{T}, \mathcal{O}, \mathcal{A}, R, P, \gamma)$, where we write the team set as $\mathcal{T} = \{T_1, T_2\}$, where $T_1$ is a finite set of players playing cooperatively against an adversary team denoted by $T_2$. Let $\mathcal{O} = \mathcal{O}_1 \times \mathcal{O}_2$ be the product of locaobservation spaces of two teams, namely the joint observation space, where $\mathcal{O}_1 = \times_{i=1}^{n_1} O_{1,i}$ and $\mathcal{O}_2 = \times_{j=1}^{n_2} O_{2,j}$ denote the product of local observation spaces of the players in team $T_1$ and $T_2$, namely team's joint observation space. $O_{1,i}, O_{2,j}$ is the local observation spaces of players $i \in T_1$ and $j \in T_2$. $\mathcal{A} = \mathcal{A}_1 \times \mathcal{A}_2$ is the product of action spaces of two teams, namely the joint action space, where $\mathcal{A}_1 = \times_{i=1}^{n_1} A_{1,i}$ and $\mathcal{A}_2 = \times_{j=1}^{n_2} A_{2,j}$ denote the product of action space of players in team $T_1$ and $T_2$, namely team's joint action space. $A_{k,i}$ is the action spaces of players $i \in T_k \in \mathcal{T}$. We define the joint action of team $T_k$ as $\boldsymbol{a}_k = (a_k^1, ...a_k^{n_k}) \in \mathcal{A}_k$. A team strategy is a vector of individual strategies of team players, denoted by $\vec{\boldsymbol{\pi}}_k = (\pi_{k,1}, ..., \pi_{k,n_1})$, where $\pi_{k,i} \in \Pi_{k,i} : O_{k,i} \to \Delta A_{k,i}$ is an individual strategy of player $i \in T_k$. $R$ is a pair of reward functions $(R_1, R_2)$, where we use the notation of $R_t : \mathcal{O} \times \mathcal{A} \to [-R_{\max}, R_{\max}], t \in \{1, 2\}$ to represent the reward function of two teams. Note that players within the same team share the team reward with $R_{k,1} = R_{k,2} = \cdots = R_{k,n_k} = R_k/n_k$, and the rewards of two teams sum to zero $R_1 + R_2 = 0$. Let $P : \mathcal{O} \times \mathcal{A} \times \mathcal{O} \to \mathbb{R}$ be the transition probability function, and $\gamma \in [0, 1)$. The transition probabilily function $P$, team policy $\vec{\boldsymbol{\pi}}_1$, opponent team policy $\vec{\boldsymbol{\pi}}_2$, and the initial observation distribution $d$, induce a marginal observation distribution at time $t$, denoted by $\rho_{\vec{\boldsymbol{\pi}}_1, \vec{\boldsymbol{\pi}}_2}^t$. At time step $t \in \mathbb{R}$, team $T_k$ observes its local observations $\boldsymbol{o}_{k,t} \in \mathcal{O}_k$ ($\boldsymbol{o}_{k,t} = (o_{k,t}^1, ..., o_{k,t}^{n_k})$ is the "joint" observations) and take team joint actions $\boldsymbol{a}_{k,t} \in \mathcal{A}_k$ according to its policy $\vec{\boldsymbol{\pi}}_k$. At each time step, two teams take actions *simultaneously* based on their observations with no sequential dependency. At the end of each time step, team $T_k$ receives its joint reward $R_k(\boldsymbol{o}_{k,t}, \boldsymbol{a}_{k,t}, \boldsymbol{o}_{k,t}, \boldsymbol{a}_{k,t})$, and observes $\boldsymbol{o}_{k,t+1}$. Following this process infinitely long, team $T_1$ and $T_2$ earn a discounted cumulative return of $R_1^\gamma \triangleq \Sigma_{t=0}^\infty \gamma^t R_1(\boldsymbol{o}_{1,t}, \boldsymbol{a}_{1,t}, \boldsymbol{o}_{2,t}, \boldsymbol{a}_{2,t})$ and of $R_2^\gamma \triangleq \Sigma_{t=0}^\infty \gamma^t R_2(\boldsymbol{o}_{1,t}, \boldsymbol{a}_{1,t}, \boldsymbol{o}_{2,t}, \boldsymbol{a}_{2,t})$ respectively. The expected reward of the team can be written as the following function:

$$R_1(\vec{\boldsymbol{\pi}}_1, \vec{\boldsymbol{\pi}}_2) := \mathbb{E}_{\boldsymbol{o}_{1,0:\infty}, \boldsymbol{o}_{2,0:\infty} \sim \rho_{\vec{\boldsymbol{\pi}}_1, \vec{\boldsymbol{\pi}}_2}^{0:\infty}, \mathbf{a}_{1,0:\infty} \sim \vec{\boldsymbol{\pi}}_1, \mathbf{a}_{2,0:\infty} \sim \vec{\boldsymbol{\pi}}_2} \left[ \sum_{t=0}^\infty \gamma^t R_1(\boldsymbol{o}_{1,t}, \boldsymbol{a}_{1,t}, \boldsymbol{o}_{2,t}, \boldsymbol{a}_{2,t}) \right].$$

*Heterogeneous* team games are a specialized subset of two-team games where each team is composed of agents with differing characteristics and abilities. Formally, $\exists i, i' \in T_k \in \mathcal{T}$ such that player $i$ and player $i'$ perform distinct roles and are not exchangeable. Consequently, the distribution of observation space $O_{k,i}$ is different from the distribution of observation space $O_{k,i'}$. Further, $A_{k,i} \cap A_{k,i'} \neq A_{k,i} \cup A_{k,i'}$. This diversity in observation and action spaces reflects the varied roles and capabilities that agents bring to the team, causes a requirement of $\pi_{k,i} \neq \pi_{k,i'}$.

**TMECor as a Maxmin Problem.** The central solution concept in *heterogeneous* team games is the Team-Maxmin Equilibrium with correlation (TMECor) (Basilico et al., 2017; Celli & Gatti, 2018). TMECor is a Nash equilibrium where the team $T_1$ plays according to the *ex ante* coordinated strategy

---

[1]Our methods mostly apply to stochastic games including Google Research Football mentioned in Section 5.2.3. Normal-form games can be considered as special cases of stochastic games with $|\mathcal{O}| = 1$.

$\pi_1 \in \Pi_1 : \mathcal{O}_1 \rightarrow \Delta\mathcal{A}_1$ and the opponent team $T_2$ plays according to the *ex ante* coordinated strategy $\pi_2 \in \Pi_2 : \mathcal{O}_2 \rightarrow \Delta\mathcal{A}_2$. According to definition, a TMECor is reached if, for each team $T \in \mathcal{T}$, its coordinated team strategy is a *best response* to the coordinated team strategies of teams $\in \mathcal{T}\backslash T$. Upon reaching a TMECor $(\pi_1^*, \pi_2^*)$, players in both $T_1$ and $T_2$ cannot cooperatively deviate from their team strategies to obtain a higher team reward:

$$R_1(\pi_1^*, \pi_2^*) \geq R_1(\pi_1, \pi_2^*) \quad \forall \pi_1 \in \Pi_1, \tag{1a}$$

$$R_2(\pi_1^*, \pi_2^*) \geq R_2(\pi_1^*, \pi_2) \quad \forall \pi_2 \in \Pi_2. \tag{1b}$$

Define the exploitability of a pair of coordinated team strategies $(\pi_1, \pi_2)$ as $e(\pi_1, \pi_2) = R_2(\pi_1, \mathbf{BR}(\pi_1)) + R_1(\mathbf{BR}(\pi_2), \pi_2)$, where $\mathbf{BR}(\pi_1)$ is the coordinated opponent team strategy which achieves the highest reward responding to the team coordinated strategy $\pi_1$ and $\mathbf{BR}(\pi_2)$ is the coordinated team strategy that achieves the highest reward responding to the coordinated opponent team strategy $\pi_2$. A coordinated team strategy pair $(\pi_1, \pi_2)$ is a TMECor if $e(\pi_1, \pi_2) = 0$, and is an $\epsilon-$approximate TMECor if $e(\pi_1, \pi_2) \leq \epsilon$.

**Policy Space Response Oracle (PSRO)** PSRO (Lanctot et al., 2017) provides an iterative mechanism for finding a Nash equilibrium approximation in two-player zero-sum games. These algorithms work in expanding a restricted policy set $\Pi_k^r$ for each team $T_k \in \mathcal{T}$ iteratively. At each epoch, a local TMECor $\sigma = (\sigma_k, \sigma_{-k})$ is computed for a restricted game which is formed by a tuple of restricted policy sets $\Pi^r = (\Pi_k^r, \Pi_{-k}^r)$. Then, a best response to the local TMECor $\sigma_{-k}$ is computed and added to team $T_k$'s restricted policy set $\Pi_k^r = \Pi_k^r \cup \{\mathbf{BR}(\sigma_{-k})\}$. When the iteration terminates with $\{\mathbf{BR}(\sigma_{-k})\} \subseteq \Pi_k^r$ and $\{\mathbf{BR}(\sigma_k)\} \subseteq \Pi_{-k}^r$, the local TMECor $\sigma^* = (\sigma_1^*, \sigma_2^*)$ for the restricted game is approximating an TMECor in the original team game.

## 3  RELATED WORK

**Team Games as Two Player Games** To compute TMECor in *heterogeneous* team games, it is straightforward to treat each *heterogeneous* team as a single player with a joint strategy space (Carminati et al., 2022). By transforming a *heterogeneous* team game into an equivalent two-player zero-sum game (2p0s), the problem of finding a TMECor becomes equivalent to the problem of finding a Nash equilibrium in two-player zero-sum games, thus more amenable to the techniques that have been developed over the past 80 years (Robinson, 1951; McMahan et al., 2003; Zinkevich et al., 2007; Lanctot et al., 2017; McAleer et al., 2020; Liu et al., 2021; Zhou et al., 2022). Celli & Gatti (2018) propose Column Generation (CG), which designs a hybrid representation to reduce the space of the join team plans and builds a subset of jointly-reduced plans progressively to avoid enumerating the whole space. While these algorithms perform well in small to medium-scale *heterogeneous* team games, scaling them to larger games is challenging because the joint policy space of both teams grows exponentially with the increasing number of players.

**Team Games as MARL Problems** Another perspective for solving *heterogeneous* team games is to formulate it as a multiplayer cooperative challenge, e.g., considering opponent team part of the environment and modeling the problem of solving TMECor as an optimization problem, which aims to maximize the reward of team $T_1$ and find an optimal *ex ante* correlation solutions for *heterogeneous* players in $T_1$. To achieve this goal, various Multi-Agent Reinforcement Learning (MARL) algorithms (Yu et al., 2022; Kuba et al., 2022; Wen et al., 2022; Wang et al., 2023) have been proposed. While these algorithms achieve remarkable performance in games like StarCraft II, they suffer from unsteady performance when applied to real-world scenarios, where diverse opponent teams are encountered (see results in Table 2).

**Team Games as Mixed Cooperative-Competitive Games** To overcome the above challenges, researchers model team games as mixed cooperative-competitive games and integrate the cooperative reinforcement learning techniques with competitive frameworks like Policy Space Response Oracle (PSRO) (Lanctot et al., 2017) to solve the mixed cooperative-competitive games. For example, McAleer et al. (2023) integrate PSRO with a homogeneous-agent based cooperative algorithms, iteratively constructing a population of shared policies to find an approximate TMECor. However, it requires players in both teams to be homogeneous, and cannot converge when applied to *heterogeneous* team games as shown in Section 4.1 and Section 4.2.

For further discussion about the technical details, please refer to Appendix D.

# 4 COMPUTING TMECOR IN HETEROGENEOUS TEAM GAMES

We focus on the problem of computing a global TMECor in *heterogeneous* team games. One of the most promising algorithms that can scale to large team games is Team PSRO (McAleer et al., 2023). However, Team PSRO heavily relies on a policy sharing mechanism for team coordination, which causes Team PSRO to be trapped into a sub-optimal *ex ante* equilibrium with significantly higher exploitability and never converges to the global TMECor in *heterogeneous* team games. In this section, we analyze and formulate the convergence problem that Team PSRO encounters in *heterogeneous* team games, and propose the first PSRO framework for *heterogeneous* team games to address the convergence issue. We analyze the reasons that cause the convergence issue of Team PSRO in *heterogeneous* team games in Section 4.1; then we formulate the convergence issue in general *heterogeneous* team games in Section 4.2. To address the convergence problem and find a global TMECor in *heterogeneous* team games, we maintain *heterogeneous* policies for teammates and propose a mechanism to reduce the high complexity brought by optimizing over multiple players' policy spaces in Section 4.3. Inspired by this, we propose a general framework named *heterogeneous* PSRO (H-PSRO) for *heterogeneous* team games in Section 4.4 and prove that H-PSRO achieves lower exploitability than Team PSRO in *heterogeneous* team games.

## 4.1 INSUFFICIENT EQUILIBRIUM EXPRESSIVE ABILITY OF NON-HETEROGENEOUS ALGORITHMS

Team PSRO relies on a policy sharing mechanism for team correlation. That is, players in team $T_k \in \mathcal{T}$ share a policy $\pi_{k,\text{share}}$, which forms a team policy $\vec{\pi}_{k,\text{share}} = \{\pi_{k,\text{share}}, \ldots, \pi_{k,\text{share}}\}$. We define the space of team policy $\vec{\pi}_{k,\text{share}}$ as $\mathbf{\Pi}_{k,\text{share}}$. Team PSRO iteratively expands a restricted policy set $\mathbf{\Pi}^r_{k,\text{share}}$ by computing a best response to the meta policy $\sigma_{-k}$ and adding it to the restricted policy set $\mathbf{\Pi}^r_{k,\text{share}} = \mathbf{\Pi}^r_{k,\text{share}} \cup \{\mathbf{BR}_{k,\text{share}}(\sigma_{-k})\}$, where the Best Response Oracle under a policy sharing based correlation is defined as $\mathbf{BR}_{k,\text{share}} : \mathbf{\Pi}_{-k} \to \mathbf{\Pi}_{k,\text{share}}$. When $\mathbf{BR}_{k,\text{share}}(\sigma_{-k})$ already exists in $\mathbf{\Pi}^r_{k,\text{share}}, \forall T_k \in \mathcal{T}$, Team PSRO terminates with a pair of meta policies $\sigma^*_{\text{share}} = (\sigma^*_{k,\text{share}}, \sigma^*_{-k,\text{share}}) \in \Delta\mathbf{\Pi}^r_{1,\text{share}} \times \Delta\mathbf{\Pi}^r_{2,\text{share}}$. While this mechanism does not introduce additional computational complexity when the number of teammates increases, it causes insufficient policy expressive ability and insufficient equilibrium expressive ability in *heterogeneous* team games.

**Example 1.** Let us consider the *heterogeneous* team game Team Rock-Paper-Scissors shown in Figure 1. In team RPS, TMECor is reached when both teams choose Rock, Paper, and Scissors with equal probability. Let the shared policies be $\pi_{1,\text{share}} := (x, 1 - x)$ and $\pi_{2,\text{share}} := (y, 1 - y)$. Team PSRO maintains two populations of shared policies denoted by $\Pi^r_{1,\text{share}}$ and $\Pi^r_{2,\text{share}}$. Initially, $\Pi^r_{1,\text{share}} = \{\pi^1_{1,\text{share}}\}$ with $\pi^1_{1,\text{share}} = (1,0)$ representing a team policy of Rock, $\Pi^r_{2,\text{share}} = \{\pi^1_{2,\text{share}}\}$ with $\pi^1_{2,\text{share}} = (1,0)$ representing an opponent team policy of Rock. To expand the population $\Pi^r_{1,\text{share}}$, the Best Response to meta policy $\pi^1_{2,\text{share}}$ of opponent team $T_2$, a Paper policy, should be added to $\Pi^r_{1,\text{share}}$. However, the Paper policy cannot be represented in the form of the shared policy. This is because team $T_1$ makes a Paper decision with a probability of $1.0$ if and only if the player $\mathsf{M_1}$ chooses action $a$ with a probability of $1.0$ ($x = 1.0$) and the player $\mathsf{M_2}$ chooses action $b$ with a probability of $1.0$ ($x = 0.0$), which is impossible at the same time. As shown by our experimental results in Figure 2, the Team PSRO algorithm is trapped into a Rock policy and never finds a global TMECor in Team RPS, even though such a solution exists. This example illustrates the convergence issue of Team PSRO due to the insufficient *policy expressive ability* and insufficient *equilibrium expressive ability*.

**Definition 1** The *Policy Expressive Ability* of team $T_1$ is defined as $PEA_{1,c} = \frac{|\mathbf{\Pi}_{1,c}|}{|\mathbf{\Pi}_1|} \leq 1$, where $\mathbf{\Pi}_{1,c}$ is the corresponding team policy space under a correlation method $c$, and $\mathbf{\Pi}_1$ is the entire team policy space.

**Definition 2** A TMECor is a vector of distributions over policy spaces of team $T_1$ and opponent team $T_2$. We define the set of TMECor within the joint policy space $\mathbf{S} = \mathbf{\Pi}_1 \times \mathbf{\Pi}_2$ as $\mathbf{E}$, and the set of TMECor within the corresponding joint policy space $\mathbf{S}_c = \mathbf{\Pi}_{1,c} \times \mathbf{\Pi}_{2,c}$ under a correlation method $c$ as $\mathbf{E}_c$, where $\mathbf{\Pi}_{k,c}$ is the team policy space of Team $T_k \in \mathcal{T}$ under the correlation method $c$.

**Proposition 1** In any two team games, *Policy Expressive Ability* of team $T_1$ under a policy sharing based correlation $PEA_{1,\text{share}} < 1$, and *Policy Expressive Ability* of opponent team $T_2$ under a policy sharing based correlation $PEA_{2,\text{share}} < 1$, leading to insufficient *policy expressive ability*.

For proof see Appendix A.1.

**Proposition 2** In *heterogeneous* team games, at most $\mathbf{E}_{\text{share}} \subseteq \mathbf{E}$; in some cases, $\mathbf{E}_{\text{share}} \neq \mathbf{E}$, leading to insufficient equilibrium expressive ability.

For proof see Appendix A.1.

### 4.2 CONVERGENCE ISSUE OF NON-HETEROGENEOUS ALGORITHMS

Due to the insufficient *policy expressive ability* and *equilibrium expressive ability* caused by policy sharing, the non-heterogeneous PSRO framework encounters severe convergence issue in *heterogeneous* team games. There are two primary reasons. Firstly, the homogeneous PSRO framework iteratively computes a Best Response policy within team policy space $\mathbf{\Pi}_{1,\text{share}}$ and $\mathbf{\Pi}_{2,\text{share}}$, and terminates if and only if Best Response policies of team $T_k$ already exist in $\mathbf{\Pi}_{k,\text{share}}^r, \forall T_k \in \mathcal{T}$. When the iteration terminates, it does not mean convergence to a TMECor in the original game because it is highly possible that the Best Response policy is in the space $\mathbf{\Pi}_k \backslash \mathbf{\Pi}_{k,\text{share}}$ because the policy expressive ability $PEA_1 < 1$ and $PEA_2 < 1$ (Proposition 1). Secondly, due to the insufficient *equilibrium expressive ability*, the homogeneous PSRO framework can only converge to a sub-optimal TMECor with no guarantee that deviating to policies within $\mathbf{\Pi}_1 \backslash \mathbf{\Pi}_{1,\text{share}}$ (or $\mathbf{\Pi}_2 \backslash \mathbf{\Pi}_{2,\text{share}}$) will not decrease (or increase) the team reward $R_1$.

**Proposition 3** In heterogeneous team games, the homogeneous PSRO framework is trapped into a sub-optimal TMECor within a subset of joint policy space $\mathbf{S}_{\text{share}} \subsetneqq \mathbf{S}$.

For proof see Appendix A.1.

### 4.3 THEOREM FOR CORRELATING HETEROGENEOUS POLICIES

To tackle the convergence issues described in Section 4.2, which is caused by insufficient *policy expressive ability* and *equilibrium expressive ability* under a policy sharing based correlation, one straightforward idea is to parameterize *heterogeneous* policies for teammates. For example, we can parameterize $\pi_{1,i}$ by $\vartheta_i$, which, together with other agents in team $T_1$, forms a joint team policy $\vec{\pi}_1$ parameterized by $\boldsymbol{\vartheta}_1 = (\vartheta_1, \ldots, \vartheta_{n_1})$. We prove that a global TMECor can be achieved with the *heterogeneously* parameterized policies.

**Theorem 1** The joint policy space with *heterogeneous* policies under PSRO framework is equal to $\mathbf{S}$, therefore enabling the PSRO framework to achieve a global *TMECor*.

For proof see Appendix A.2.

However, the *heterogeneous* policies bring in new difficulties for finding the optimal *ex ante* correlated solution. Optimizing over multiple players' policy space simultaneously is significantly harder than optimizing over a single shared policy space (e.g., $\Delta_2^P$-complete) (Zhang et al., 2022b), and optimizing over the correlated team policy space consisting of *heterogeneous* policies makes the optimization space grow exponentially with the increasing number of players. To find an optimal *ex ante* correlation solution for *heterogeneous* policies without introducing additional complexity, we propose a sequential correlation method for the Best Response Oracle (BRO) for computing a best response *ex ante* correlation strategy for a given opposing team's strategy. The proposed sequential correlation method is based on the observation in Lemma 1 (see Appendix A.1).

Lemma 1 confirms that a sequential update is an effective approach for sequential BRO to search for the direction of *ex ante* coordination improvement (i.e., joint actions with positive advantage values) in two-team games given the opposing team's policy. That is, agents take actions sequentially by following an arbitrary order $\vec{i}_{1:n_1} = (i_1, \ldots, i_{n_1}) = T_1$ (or $\vec{j}_{1:n_2} = (i_1, \ldots, i_{n_2}) = T_2$). Let agent $i_1 \in T_1$ (or $j_1 \in T_2$) take action $a_{1,i_1}$ (or $a_{2,j_1}$) such that the value of the advantage function of taking action $a_{1,i_1}$ (or the value of the advantage function of taking action $a_{2,j_1}$) is positive, and then, for the remaining $m = 2, \ldots, n_1$ (or $n_2$), each agent $i_m \in T_1$ (or $j_m \in T_2$) takes an action $a_{1,i_m}$ (or $a_{2,i_m}$) such that the advantage function value of taking action $a_{1,i_m}$ (or $a_{2,j_m}$) conditioned on the joint action $\vec{a}_{1,i_1:i_{m-1}}$ (or $\vec{a}_{2,j_1:j_{m-1}}$) is positive. For the induced joint action $\vec{a}_1$ (or $\vec{a}_2$), as shown in Lemma 1, the advantage function value of taking action $\vec{a}_1$ (or $\vec{a}_2$) is positive, thus the

coordination performance of both team $T_1$ and opponent team $T_2$ is guaranteed to improve. Such a sequential correlation offers a solution for monotonic improvement towards optimal *ex ante* team correlations with a linearly growing optimization space with the increasing number of teammates. The detailed process of such a sequential BRO is summarized in Algorithm 2 in Appendix E.

**Lemma 2** In any two team games, $\mathbf{\Pi}_{k,\text{share}} \subsetneq \mathbf{\Pi}_{k,\text{seq}}$ holds for all $T_k \in \mathcal{T}$, where $\mathbf{\Pi}_{k,\text{share}}$ is the policy search space of team $T_k$ with a policy sharing based BRO, and $\mathbf{\Pi}_{k,\text{seq}}$ is the policy search space of team $T_k$ with sequential BRO.

For proof see Appendix A.1. Lemma 2 shows that the search space of sequential BRO contains and is larger than the search space of the policy sharing based BRO. With lemma 2, we claim a theorem of superior *ex ante* correlation property of the sequential BRO.

**Theorem 2** Given an opponent team policy $\boldsymbol{\pi_2} \in \mathbf{\Pi_2}$ (or a team policy $\boldsymbol{\pi_1} \in \mathbf{\Pi_1}$), the sequential BRO can achieve better *ex ante* team coordination than the policy sharing based BRO with $R_1(\mathbf{BR}_{1,\text{seq}}(\boldsymbol{\pi_2}), \boldsymbol{\pi_2}) \geq R_1(\mathbf{BR}_{1,\text{share}}(\boldsymbol{\pi_2}), \boldsymbol{\pi_2})$ and $R_2(\boldsymbol{\pi_1}, \mathbf{BR}_{2,\text{seq}}(\boldsymbol{\pi_1})) \geq R_2(\boldsymbol{\pi_1}, \mathbf{BR}_{2,\text{share}}(\boldsymbol{\pi_1}))$, where $\mathbf{BR}_{k,\text{share}} : \mathbf{\Pi}_{-k} \to \mathbf{\Pi}_{k,\text{share}}$ is the policy sharing based BRO of team $T_k \in \mathcal{T}$, and $\mathbf{BR}_{k,\text{seq}} : \mathbf{\Pi}_{-k} \to \mathbf{\Pi}_{k,\text{seq}}$ is the sequential BRO of team $T_k \in \mathcal{T}$. In some cases, $R_1(\mathbf{BR}_{1,\text{seq}}(\boldsymbol{\pi_2}), \boldsymbol{\pi_2}) > R_1(\mathbf{BR}_{1,\text{share}}(\boldsymbol{\pi_2}), \boldsymbol{\pi_2})$ and $R_2(\boldsymbol{\pi_1}, \mathbf{BR}_{2,\text{seq}}(\boldsymbol{\pi_1})) > R_2(\boldsymbol{\pi_1}, \mathbf{BR}_{2,\text{share}}(\boldsymbol{\pi_1}))$ hold.

For proof see Appendix A.3. As a natural result of lemma 2, the sequential BRO can find an *ex ante* coordinated team strategy with higher team reward than the policy sharing based BRO, as proved in Theorem 2. This theorem provides an idea about how sequentially correlated heterogeneous policies can have superior performance than the correlated shared policies. Note that this property holds with no requirement on the specific order by which teammates make their updates.

---

**Algorithm 1:** Heterogeneous PSRO

1 **input :** initial policy sets for teams $\mathbf{\Pi}^r_{1,\text{hete}}, \mathbf{\Pi}^r_{2,\text{hete}}$
2 **for** $t = \{1, 2, \cdots T\}$ **do**
3      Compute utilities $U_{1\times 2}$ for each joint $\vec{\pi}_{1,\text{hete}} \in \mathbf{\Pi}^r_{1,\text{hete}}, \vec{\pi}_{2,\text{hete}} \in \mathbf{\Pi}^r_{2,\text{hete}}$.
4      $(\sigma_{1,\text{seq}}, \sigma_{2,\text{seq}}) = \text{MetaSolver}(U_{1\times 2})$
5      **for** *team* $T_k \in \mathcal{T}$ **do**
6          $\vec{\pi}_{k,\text{hete}} = \text{SequentialBRO}(\sigma_{-k,\text{seq}}, \mathbf{\Pi}^r_{-k,\text{hete}})$. `// see Algo. 2 in Appx. E.`
7          $\mathbf{\Pi}^r_{k,\text{hete}} = \mathbf{\Pi}^r_{k,\text{hete}} \cup \{\vec{\pi}_{k,\text{hete}}\}$.
8 **Output :** local TMECor $(\sigma_{1,\text{seq}}, \sigma_{2,\text{seq}})$ in the current restricted game

---

## 4.4 A GENERAL FRAMEWORK FOR HETEROGENEOUS TEAM GAMES

Inspired by the results in Section 4.3, we introduce a general framework, Heterogeneous PSRO (H-PSRO). H-PSRO integrates the sequential correlation mechanism derived from lemma 1 into an iterative procedure and serves as the first PSRO framework for *heterogeneous* team games. We prove that H-PSRO can achieve lower exploitability than the homogeneous PSRO framework in *heterogeneous* team games. H-PSRO iteratively expands a restricted set of team policies $\mathbf{\Pi}^r_{k,\text{hete}}$ for team $T_k \in \mathcal{T}$, where each team policy consists of coordinated *heterogeneous* policies denoted by $\vec{\pi}_{k,\text{hete}} = \{\pi_{k,1}, \pi_{k,2}, \ldots, \pi_{k,n_k}\}$, where *heterogeneous* policies $\pi_{k,1}, \pi_{k,2}, \ldots, \pi_{k,n_k}$ are played independently and we define the space of team policy $\vec{\pi}_{k,\text{hete}}$ as $\mathbf{\Pi}_{k,\text{hete}} \subseteq \mathbf{\Pi}_k$. As proved in Theorem 1, the joint policy space under the PSRO framework $\mathbf{S}_{\text{hete}} = \Delta\mathbf{\Pi}^r_{1,\text{hete}} \times \Delta\mathbf{\Pi}^r_{2,\text{hete}} = \mathbf{S}$, thus mitigating the problem of insufficient equilibrium expressive ability in Proposition 2.

H-PSRO starts with randomly initialized team policies $\vec{\pi}^1_{1,\text{hete}}, \vec{\pi}^1_{2,\text{hete}}$, and the restricted sets of team policies and opponent team policies are $\mathbf{\Pi}^r_{1,\text{hete}} = \{\vec{\pi}^1_{1,\text{hete}}\}, \mathbf{\Pi}^r_{2,\text{hete}} = \{\vec{\pi}^1_{1,\text{hete}}\}$. Consider the restricted game where the team policy space is $\mathbf{\Pi}^r_{1,\text{hete}}$ and the opponent team policy space is $\mathbf{\Pi}^r_{2,\text{hete}}$. We denote the payoff matrix of this restricted game as $U_{1\times 2}$. If the game is symmetric, we also have a joint population $\mathbf{\Pi}_{1+2} = \mathbf{\Pi}^r_{1,\text{hete}} \cup \mathbf{\Pi}^r_{2,\text{hete}}$, and the corresponding payoff matrix is denoted as $U_{1+2} = U_{(1+2)\times(1+2)}$. In each iteration, H-PSRO expands the restricted policy set $\mathbf{\Pi}^r_{k,\text{hete}}, T_k \in \mathcal{T}$ by computing a Best Response policy with sequential BRO denoted by $\mathbf{BR}_{k,\text{seq}} : \mathbf{\Pi}_{-k} \to \mathbf{\Pi}_{k,\text{seq}}$ against the meta policy $\sigma_{-k,\text{seq}}$ of opposing team, which is a local TMECor probability over the restricted policy set $\mathbf{\Pi}^r_{-k,\text{hete}}$, and adding the best response policy to the restricted policy set

$\mathbf{\Pi}_{k,\text{hete}}^r = \mathbf{\Pi}_{k,\text{hete}}^r \cup \{\mathbf{BR}_{k,\text{seq}}(\sigma_{-k,\text{seq}})\}$. The detailed procedure of sequential BRO is shown in Algorithm 2 in Appendix E. Theorem 2 proves that the sequential BRO can achieve better *ex ante* team coordination than the policy sharing based BRO in the homogeneous PSRO framework. At the end of each iteration, the payoff matrix $U_{\mathbf{1} \times \mathbf{2}}$ (or $U_{(\mathbf{1+2}) \times (\mathbf{1+2})}$) is updated by game simulations. H-PSRO terminates with a local TMECor $\sigma_{\text{seq}}^* = (\sigma_{\mathbf{1},\text{seq}}^*, \sigma_{\mathbf{2},\text{seq}}^*) \in \Delta \mathbf{\Pi}_{\mathbf{1},\text{hete}}^r \times \mathbf{\Pi}_{\mathbf{2},\text{hete}}^r$ after convergence or a fixed number of training steps. The process is summarized in Algorithm 1.

**Theorem 3** In *heterogeneous* team games, H-PSRO achieves lower exploitability than Team PSRO. Formally, $e(\sigma_{\mathbf{1},\text{seq}}^*, \sigma_{\mathbf{2},\text{seq}}^*) \le e(\sigma_{\mathbf{1},\text{share}}^*, \sigma_{\mathbf{2},\text{share}}^*)$.

For proof see Appendix A.4. The superior *ex ante* correlation property (Theorem 2), achieved through the sequential updates and sequential BRO, provided us with a guarantee on the better convergence to the global TMECor, as shown in Theorem 3. The proof is finalised by excluding a possibility that the search space of sequential BRO $\mathbf{\Pi}_{k,\text{seq}}$ cannot cover all pure team strategies for any team $T_k \in \mathcal{T}$.

## 5 EXPERIMENTS

The main purpose of the experiments is to compare H-PSRO with existing state-of-the-art PSRO variants in terms of approximating a full game TMECor. The baseline methods include Team PSRO (McAleer et al., 2023), PSRO (Lanctot et al., 2017), Indep-PSRO[2], Self Play, Fictitious Self Play (FSP) (Heinrich et al., 2015). The benchmarks consist of single-state heterogeneous team games (Team Rock-Paper-Scissors and Matrix heterogeneous Team games) and complex stochastic heterogeneous team games (Competitive StarCraft and Google Research Football). For Team Rock-Paper-Scissors, Matrix heterogeneous Team games and Competitive StarCraft, we report the exploitability of the meta TMECor or learning trajectories through the training process. For Google Research Football where the exact exploitability is intractable, we report the performance of the final strategies. First, we analyze the empirical convergence performance of H-PSRO and several baselines in a case study of Team Rock-Paper-Scissors, which is an extended heterogeneous team games of the classic two-player zero-sum game Rock-Paper-Scissors. In addition, we illustrate how the performance evolves for each method using the Competitive StarCraft Benchmark in Section 5.2.2 and the MAgent game (Zheng et al., 2018; Terry et al., 2020) in Appendix B.1, where H-PSRO is more effective at approximating a TMECor with the enlarging task scales. An ablation study on relative performance against state-of-the-art MARL algorithms of H-PSRO in Appendix B.2 reveals that, with different MARL opponent strategies, H-PSRO exhibits superior win rate and more steady performance. The competitive videos against state-of-the-art MARL algorithms are available at https://sites.google.com/view/h-psro-2024/h-psro.

### 5.1 A CASE STUDY: TEAM ROCK-PAPER-SCISSORS

We analyze the convergence property of H-PSRO and other baselines in Team Rock-Paper-Scissors (team RPS), which extends the classic 2-player zero-sum game Rock-Paper-Scissors to a 4-player heterogeneous team setting (see details in Example 1). This task requires agents in the same team to cooperatively choose Rock, Paper, Scissors to compete against the opposing team. Clearly, this game has a unique TMECor where the team chooses Rock, Paper, Scissors with equal probability. However, as analyzed in Example 1, Team PSRO fails to find such a TMECor because the insufficient policy expressive ability caused by the policy sharing based correlation makes the equilibrium set $\mathbf{E}_{\text{share}} \ne \mathbf{E}$.

We visualize the trajectories of Self Play (SP), Fictitious Self Play (FSP), Team PSRO, and H-PSRO in Team RPS, and the learning dynamics are shown in Figure 2. The orange star in each subfigure is the TMECor of the team RPS game. The black lines in SP subfigure are the traces of the training policies and in FSP subfigure are the traces of their time-averaged policies. In Team PSRO and H-PSRO subfigures are the mixed policies of current populations. As depicted in the figure, SP transitions sequentially from Rock to Paper, to Scissors, and then back to Rock, getting perpetually entrapped in a non-transitive cycle, FSP cycles aroung the TMECor and sees the similar non-transitive cycle, Team PSRO, with insufficient policy expressive ability, cycles around the Rock policy permanently, and H-PSRO quickly converges to the TMECor.

---

[2]In Indep-PSRO, teammates have heterogeneous policies and optimize their policies simultaneously with no optimal guarantee.

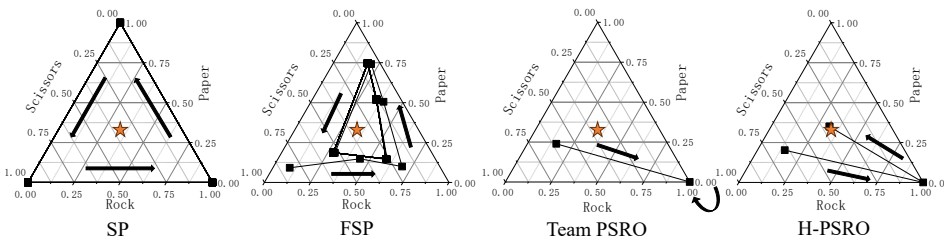

Figure 2: Trajectories of SP, FSP, Team PSRO and H-PSRO in Team RPS game (Example 1). H-PSRO shows superior convergence to the global TMECor due to the sufficient policy expressive ability of heterogeneous policies and the corresponding full equilibrium expressiveness under the heterogeneous PSRO framework (see Theorem 1).

## 5.2 HETEROGENEOUS TEAM GAMES

In this section, we study the impacts of the insufficient policy expressive ability in different *heterogeneous* team games including a matrix heterogeneous team game, competitive StarCraft, and Google Research Football.

### 5.2.1 MATRIX HETEROGENEOUS TEAM GAMES

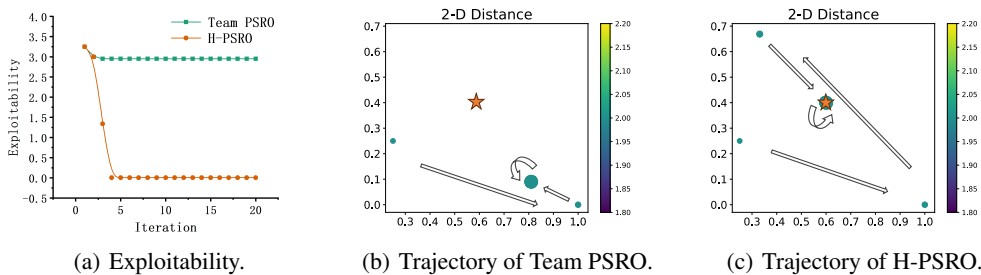

Figure 3: Performance of H-PSRO and Team PSRO in a typical Matrix Heterogeneous Team Game.

We conduct our matrix experiment on a carefully designed heterogeneous team game, which involves two teams: $T_1 = \{M_1, M_2\}$ and $T_2 = \{O_1, O_2\}$ with joint team action spaces $\mathcal{A}_1 = \{0, 1\} \times \{0, 2\}$ and $\mathcal{A}_2 = \{0, 1\} \times \{0, 3\}$, and the reward structure of this game is defined in Eq (5). The heterogeneity lies in the different action spaces of team players in $T_1$ and $T_2$. The global TMECor in this game requires team $T_1$ to take joint action $(0, 0)$ with probability $0.6$, $(0, 2)$ with probability $0.4$, and all other joint actions with probability $0$, and requires opponent team $T_2$ to take joint action $(0, 0)$ with probability $0.4$, $(1, 0)$ with probability $0.6$, and all joint other actions with probability $0$. We visualize the trajectory of the 8-dimensional joint policies of two teams in a compressed 2D space in Figure 3(b) and Figure 3(c) in order to compare the convergence properties of H-PSRO and Team PSRO. The results show that Team PSRO gets stuck in a sub-optimal point with $\sigma_{1,\text{share}} = (0.81, 0.09, 0.09, 0.01)$ and $\sigma_{2,\text{share}} = (1., 0., 0., 0.)$, where $R_1(\sigma_{1,\text{share}}, \mathbf{BR}(\sigma_{1,\text{share}})) \approx 4.0$ and $R_2(\mathbf{BR}(\sigma_{2,\text{share}}), \sigma_{2,\text{share}}) \approx -1.05$, leading to exploitability $\approx 2.95$ (see Figure 3(a)). In contrast, H-PSRO approximates the global TMECor with exploitability $< 10^{-6}$. The exploitability outcomes nicely align with Theorem 3, demonstrating H-PSRO's superior ability to explore sufficient policy spaces, and to approximate the global TMECor in heterogeneous team games.

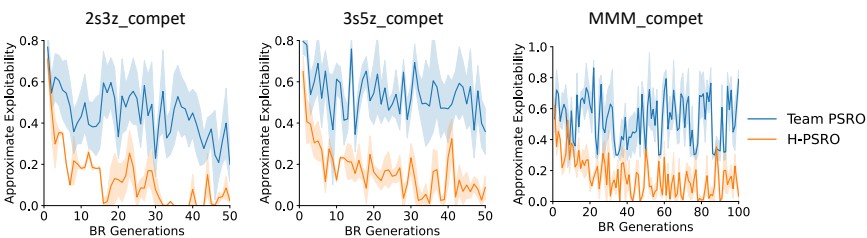

Figure 4: Exploitability of H-PSRO and Team PSRO in the Competitive StarCraft Benchmark.

### 5.2.2 COMPETITIVE STARCRAFT

Competitive StarCraft (Leroy et al., 2022) is a variant of the classical StarCraft Multi-Agent Challenge (SMAC, Samvelyan et al., 2019), which allows both team $T_1$ and opponent team $T_2$ to control their actions and naturally serves a heterogeneous team game benchmark. The Competitive StarCraft provides a set of StarCraft II maps to evaluate the effectiveness of H-PSRO. These maps feature a team of mostly heterogeneous ally units that aim to defeat a team of heterogeneous enemy units, and challenges algorithms to ensure internal team cooperation while learning robust equilibrium strategies that are capable of facing diverse opponent team strategies. We compare H-PSRO and Team PSRO in Competitive StarCraft maps with different scales, including 2s3z_compet, 3s5z_compet, and MMM_compet (see details in Table 2), and the exploitability results are shown in Figure 4.

These results show that H-PSRO achieves superior convergence with an exploitability of approximate 0 across different tasks while Team PSRO converges to strategies with significant higher exploitability. With the task scale increases, we see a larger exploitability gap between H-PSRO and Team PSRO, which hints that the impact of insufficient policy expressive ability expands with the game scales. Also, we can find that H-PSRO consistently converges at a faster speed than Team PSRO.

### 5.2.3 GOOGLE RESEARCH FOOTBALL

Google Research Football environment is a simulation environment for real-world football games, where each team consists of diverse players such as forwards, midfielders, defenders, and goalkeepers. To further demonstrate the effectiveness of H-PSRO and the impact of insufficient policy expressive ability in more complex heterogeneous environment, we utilized Google Research Football (GRF) (Kurach et al., 2020) as our benchmark. Specifically, we conducted training and evaluation of H-PSRO and baseline algorithms on the full 5-vs-5 game in GRF based on the benchmark (Song et al., 2023).

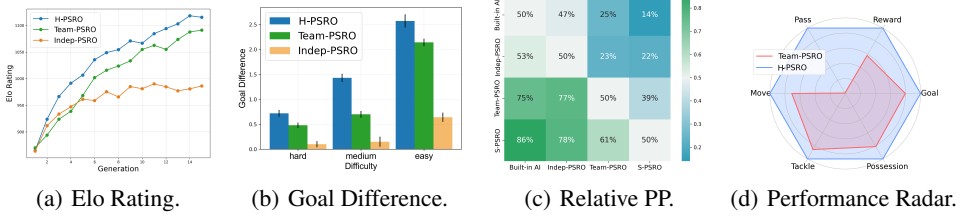

|        (a) Elo Rating.        |        (b) Goal Difference.        |        (c) Relative PP.        |        (d) Performance Radar.        |

Figure 5: Performance of H-PSRO, Team PSRO and Indep-PSRO in Google Research Football.

Because the game is too complex, it is impossible to exactly calculate or approximately estimate the exploitability of an equilibrium policy. As an alternative approach, we evaluate H-PSRO policy and other baseline outcomes by playing against a collection of unseen benchmark policies and compare their performance. The results are shown in Figure 5(a), Figure 5(b) and Figure 5(c), where H-PSRO achieves superior *Elo rating* (Elo & Sloan, 1978), largest average *goal difference* against all difficulty levels of built-in AI than Team PSRO, and highest relative population performance (*Relative PP*) (Vinyals et al., 2019). Furthermore, H-PSRO also significantly outperforms Team PSRO in terms of specific behaviours, including cooperative behaviour Pass, Tackle, Move, and goal possession behaviours Reward, Move, and Reward, as shown in Figure 5(d). These results show that H-PSRO can scale to complex heterogeneous team games, while Team PSRO's performance is hindered by the severely insufficient policy expressive ability.

## 6 CONCLUSION

In this work, we introduced Heterogeneous-PSRO (H-PSRO), a framework addressing limitations in computing *ex ante* equilibria in *heterogeneous* team games. By incorporating a sequential correlation mechanism, H-PSRO expands policy expressiveness in heterogeneous settings without exponentially increasing computational complexity. We demonstrated that this leads to monotonic improvements in team coordination, overcoming convergence issues in Team PSRO. Our theoretical analysis and empirical results confirm that H-PSRO achieves lower exploitability in both matrix games and large-scale benchmarks, outperforming homogeneous baselines in both scenarios. However, the reliance on sequential optimization may limit its efficiency in highly complex games with a large number of agents. Additionally, the scalability of the method to games with continuous action spaces requires further investigation. Despite these limitations, H-PSRO shows strong potential to scale and deliver robust solutions in complex multiplayer environments.

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

## A  PROOF

### A.1  USEFUL LEMMAS

**Proposition 1** In any two team games, *Policy Expressive Ability* of team $T_1$ under a policy sharing based correlation $PEA_{1,\text{share}} < 1$, and *Policy Expressive Ability* of opponent team $T_2$ under a policy sharing based correlation $PEA_{2,\text{share}} < 1$, leading to insufficient *policy expressive ability*.

*Proof.* Consider any two team games where each team contains at least two players, at any observations, joint policy space $\mathbf{\Pi}_{1,\text{share}} = \{x_1 \text{ for } a_1, x_2 \text{ for } a_2 | x_1 \in [0,1], x_2 \in [0,1], x_1 + x_2 \le 1\}$, and $\mathbf{\Pi}_{2,\text{share}} = \{y_1 \text{ for } b_1, y_2 \text{ for } b_2 | y_1 \in [0,1], y_2 \in [0,1], y_1 + y_2 \le 1\}$. The $PEA_{1,\text{share}} < 1$ because any pure policies where action $a_1$ is played by one player and $a_2$ is played by another player requires $x_1 = 1, x_2 = 1$ to hold at the same time, which is impossible, making these pure policies $\in \mathbf{\Pi}_1 \backslash \mathbf{\Pi}_{1,\text{share}} \ne \emptyset$. Similarly, $PEA_{2,\text{share}} < 1$ because pure policies where $b_1$ is played by one player and $b_2$ is played by another player $\in \mathbf{\Pi}_2 \backslash \mathbf{\Pi}_{2,\text{share}} \ne \emptyset$. $\square$

**Proposition 2** In *heterogeneous* team games, at most $\mathbf{E}_{\text{share}} \subseteq \mathbf{E}$; in some cases, $\mathbf{E}_{\text{share}} \ne \mathbf{E}$, leading to insufficient equilibrium expressive ability.

*Proof.* According to the definition of TMECor, $\mathbf{E}_{\text{share}} \subseteq \mathbf{E}$ if and only if the following conditions hold for all $(\boldsymbol{\pi}_1^*, \boldsymbol{\pi}_2^*) \in \mathbf{E}_{\text{share}}$:

$$R_1(\boldsymbol{\pi}_1^*, \boldsymbol{\pi}_2^*) \ge R_1(\boldsymbol{\pi}_1, \boldsymbol{\pi}_2^*), \quad \forall \boldsymbol{\pi}_1 \in \mathbf{\Pi}_1 \backslash \mathbf{\Pi}_{1,\text{share}}, \tag{2a}$$

$$R_2(\boldsymbol{\pi}_1^*, \boldsymbol{\pi}_2^*) \ge R_2(\boldsymbol{\pi}_1^*, \boldsymbol{\pi}_2), \quad \forall \boldsymbol{\pi}_2 \in \mathbf{\Pi}_2 \backslash \mathbf{\Pi}_{2,\text{share}}. \tag{2b}$$

Otherwise, if $\exists (\boldsymbol{\pi}_1^*, \boldsymbol{\pi}_2^*) \in \mathbf{E}_{\text{share}}$ and $(\boldsymbol{\pi}_1^*, \boldsymbol{\pi}_2^*)$ does not satisfy the requirements of TMECor within the whole equilibrium space $\mathbf{S}$, $\mathbf{E}_{\text{share}} \ne \mathbf{E}$. In another case where $\exists (\boldsymbol{\pi}_1^*, \boldsymbol{\pi}_2^*) \in \mathbf{E}$ and $(\boldsymbol{\pi}_1^*, \boldsymbol{\pi}_2^*) \in \mathbf{S} \backslash \mathbf{S}_{\text{share}}$, the global TMECor $(\boldsymbol{\pi}_1^*, \boldsymbol{\pi}_2^*)$ does not exist in $\mathbf{E}_{\text{share}}$, also making $\mathbf{E}_{\text{share}} \ne \mathbf{E}$.

$\square$

**Proposition 3** In heterogeneous team games, the homogeneous PSRO framework is trapped into a sub-optimal TMECor within a subset of joint policy space $\mathbf{S}_{\text{share}} \subsetneqq \mathbf{S}$.

*Proof.* Firstly, the homogeneous PSRO framework iteratively computes a Best Response policy within team policy space $\mathbf{\Pi}_{1,\text{share}}$ and $\mathbf{\Pi}_{2,\text{share}}$, and terminates if and only if Best Response policies of team $T_k$ already exist in $\mathbf{\Pi}_{k,\text{share}}^r$ for all $T_k \in \mathcal{T}$. When the iteration terminates, it does not mean a convergence to a TMECor in the original game because it is highly possible that the Best Response policy is in the space $\mathbf{\Pi}_k \backslash \mathbf{\Pi}_{k,\text{share}}$ (Proposition 1).

Further, let $\sigma_{\text{share}}^* = (\sigma_{1,\text{share}}^*, \sigma_{2,\text{share}}^*)$ be a local TMECor found by the homogeneous PSRO framework, where each $\sigma_{k,\text{share}}^* \in \Delta \mathbf{\Pi}_{k,\text{share}}^r$ is the local policy for team $T_k \in \mathcal{T}$ that maximizes the

minimal expected utility of the meta policy of its opposing team $\sigma_{-k,\text{share}}$. The local policy profile $\sigma^*_{\text{share}} = (\sigma^*_{1,\text{share}}, \sigma^*_{2,\text{share}})$ satisfies the following condition:

$$R_1(\sigma^*_{1,\text{share}}, \sigma^*_{2,\text{share}}) = \max_{\sigma_{1,\text{share}} \in \Delta\Pi^r_{1,\text{share}}} \min_{\sigma_{2,\text{share}} \in \Delta\Pi^r_{2,\text{share}}} R_1(\sigma_{1,\text{share}}, \sigma_{2,\text{share}}),$$

where $\Pi^r_{k,\text{share}} \subseteq \Pi_{k,\text{share}}$ is the restricted set of team policies of $T_k \in \mathcal{T}$, and $\Delta\Pi^r_{k,\text{share}}$ is the set of distributed policies over $\Pi^r_{k,\text{share}}$. According to Proposition 1, $\Pi_{k,\text{share}} \subsetneqq \Pi_k, \forall T_k \in \mathcal{T}$, and apparently $\sigma_{k,\text{share}} \in \Pi_{k,\text{share}}$ for all $\sigma_{k,\text{share}} \in \Delta\Pi_{k,\text{share}}$. As a result, $(\sigma^*_{1,\text{share}}, \sigma^*_{2,\text{share}})$ is a local TMECor within a subset of joint policy space $\Pi_{1,\text{share}} \times \Pi_{2,\text{share}}$, and there is no guarantee that:

$$R_1(\sigma^*_{1,\text{share}}, \sigma^*_{2,\text{share}}) \geq R_1(\sigma_{1,\text{share}}, \sigma^*_{2,\text{share}}) \quad \forall \sigma_{1,\text{share}} \in \Pi_1 \backslash \Pi_{1,\text{share}}, \tag{3a}$$

$$R_2(\sigma^*_{1,\text{share}}, \sigma^*_{2,\text{share}}) \geq R_2(\sigma^*_{1,\text{share}}, \sigma_{2,\text{share}}) \quad \forall \sigma_{2,\text{share}} \in \Pi_2 \backslash \Pi_{2,\text{share}}. \tag{3b}$$

$\square$

**Lemma 1 (Teammates Advantage Decomposition)** (Kuba et al., 2022) In any two team games, given a confronting joint policy of the opposing team $\vec{\pi}_2$(or $\vec{\pi}_1$), for any observation $o_1$(or $o_2$), and any agent sequence $i_{1:m} \subseteq T_1$(or $j_{1:m} \subseteq T_2$), the value of advantage functions of taking action $a_1^{i_{1:m}} = (a_1^{i_1}, a_1^{i_2}, \ldots, a_1^{i_m})$ or $(a_2^{j_{1:m}} = (a_2^{j_1}, a_2^{j_2}, \ldots, a_2^{j_m}))$ following the joint team policy $\vec{\pi}_1$(or $\vec{\pi}_2$) is equal to the sum of the advantage function of taking action $a_{1,i_m}$ (or $a_{2,j_m}$) and taking joint action $a_1^{i_{1:m-1}}$ (or $a_2^{j_{1:m-1}}$), where an advantage function denoted by $A^{i_{1:m}}_{\vec{\pi}_1}(o_1, a_1^{i_{1:m}}|\vec{\pi}_2)$ (or $A^{j_{1:m}}_{\vec{\pi}_2}(o_2, a_2^{j_{1:m}}|\vec{\pi}_1)$) quantifies the expected gain from taking a particular action $a_1^{i_{1:m}}$ (or $a_2^{j_{1:m}}$) or following the optimized strategy relative to the policy before optimization, and an advantage function denoted by $A^{i_l}_{\vec{\pi}_1}\left(o_1, a_1^{i_{1:l-1}}, a_1^{i_l} \mid \vec{\pi}_2\right)$ (or $A^{j_l}_{\vec{\pi}_2}\left(o_2, a_2^{j_{1:l-1}}, a_2^{j_l} \mid \vec{\pi}_1\right)$) quantifies the expected gain from taking an action $a_1^{i_l}$ (or $a_2^{i_l}$) conditioned on the joint action $a_1^{i_{1:l-1}}$ (or $a_2^{j_{1:l-1}}$) that has been chosen (see details in Eq (6)). Formally,

$$A^{i_{1:m}}_{\vec{\pi}_1}\left(o_1, a_1^{i_{1:m}} \mid \vec{\pi}_2\right) = \sum_{l=1}^{m} A^{i_l}_{\vec{\pi}_1}\left(o_1, a_1^{i_{1:l-1}}, a_1^{i_l} \mid \vec{\pi}_2\right), \tag{4a}$$

$$A^{j_{1:m}}_{\vec{\pi}_2}\left(o_2, a_2^{j_{1:m}} \mid \vec{\pi}_1\right) = \sum_{l=1}^{m} A^{j_l}_{\vec{\pi}_2}\left(o_2, a_2^{j_{1:l-1}}, a_2^{j_l} \mid \vec{\pi}_1\right). \tag{4b}$$

*Proof.* By the definition of teammate advantage function and state-action value function in Appendix D.2,

$$A^{i_{1:m}}_{\vec{\pi}_k}\left(o_k, a_k^{i_{1:m}}|\vec{\pi}_{-k}\right) = Q^{i_{1:m}}_{\vec{\pi}_k}\left(o_k, a_k^{i_{1:m}}|\vec{\pi}_{-k}\right) - V_{\vec{\pi}_k}(o_k|\vec{\pi}_{-k})$$

$$= \sum_{l=1}^{m}\left[Q^{i_{1:l}}_{\vec{\pi}_k}\left(o_k, a_k^{i_{1:l}}|\vec{\pi}_{-k}\right) - Q^{i_{1:l-1}}_{\vec{\pi}_k}\left(o_k, a_k^{i_{1:l-1}}|\vec{\pi}_{-k}\right)\right]$$

$$= \sum_{l=1}^{m} A^{i_l}_{\vec{\pi}_k}\left(o_k, a_k^{i_{1:l-1}}, a_k^{i_l}|\vec{\pi}_{-k}\right), \forall T_k \in \mathcal{T}\}$$

which finishes the proof.

$\square$

**Lemma 2** In any two team games, $\Pi_{k,\text{share}} \subsetneqq \Pi_{k,\text{seq}}$ holds for all $T_k \in \mathcal{T}$, where $\Pi_{k,\text{share}}$ is the search space of team $T_k$ with a policy sharing based BRO, and $\Pi_{k,\text{seq}}$ is the search space of team $T_k$ under with sequential BRO.

*Proof.* The BRO of team $T_k \in \mathcal{T}$ under a policy sharing based correlation is defined as $\mathbf{BR}_{k,\text{share}} : \Pi_{-k} \to \Pi_{k,\text{share}}$, and the sequential BRO of team $T_k \in \mathcal{T}$ is defined as $\mathbf{BR}_{k,\text{seq}} : \Pi_{-k} \to \Pi_{k,\text{seq}}$. Given an opponent team strategy $\pi_{-k} \in \Pi_{-k}$, the policy sharing based BRO computes a best response of coordinated shared strategy $\mathbf{BR}_{k,\text{share}}(\pi_{-k}) = \arg\max_{\vec{\pi}_{k,\text{share}} \in \Pi_{k,\text{share}}} R_k(\vec{\pi}_{k,\text{share}}, \pi_{-k})$,

while the sequential BRO computes a best response of coordinated heterogeneous strategy $\mathbf{BR}_{k,\text{seq}}(\boldsymbol{\pi}_{-k}) = \arg\max_{\vec{\boldsymbol{\pi}}_{k,\text{hete}} \in \boldsymbol{\Pi}_{k,\text{seq}}} R_k(\vec{\boldsymbol{\pi}}_{k,\text{hete}}, \boldsymbol{\pi}_{-k})$. With the sequential update mechanism in Algorithm 2 in Appendix E, the search space $\boldsymbol{\Pi}_{k,\text{seq}} = \{(\pi_{k,1}, \pi_{k,2}, \dots, \pi_{k,n_k}) | \pi_{k,1} \in \Pi_{k,1}, \pi_{k,2} \in \Pi_{k,2}, \dots, \pi_{k,n_k} \in \Pi_{k,n_k}\}$. The search space $\boldsymbol{\Pi}_{k,\text{share}} = \{(\pi_{k,1}, \pi_{k,2}, \dots, \pi_{k,n_k}) | \pi_{k,1} = \pi_{k,2} = \dots = \pi_{k,n_k}, \pi_{k,1} \in \Pi_{k,1}, \pi_{k,2} \in \Pi_{k,2}, \dots, \pi_{k,n_k} \in \Pi_{k,n_k}\}$ is a subset of $\boldsymbol{\Pi}_{k,\text{seq}}$. On the other hand, $\exists \vec{\boldsymbol{\pi}}_{k,\text{hete}} \in \boldsymbol{\Pi}_{k,\text{seq}}$ and $\vec{\boldsymbol{\pi}}_{k,\text{hete}} \notin \boldsymbol{\Pi}_{k,\text{share}}$, making $\boldsymbol{\Pi}_{k,\text{share}} \neq \boldsymbol{\Pi}_{k,\text{seq}}$. As a result, $\boldsymbol{\Pi}_{k,\text{share}} \subsetneqq \boldsymbol{\Pi}_{k,\text{seq}}$.

$\square$

### A.2 PROOF OF SUFFICIENT EQUILIBRIUM EXPRESSIVE ABILITY OF HETEROGENEOUS TEAM POLICIES

**Theorem 1** The joint policy space with *heterogeneous* policies under PSRO framework is equal to **S**, therefore enabling the PSRO framework to achieve a global *TMECor*.

*Proof.* With heterogeneous policies, for example, $\vec{\boldsymbol{\pi}}_{1,\text{hete}} = (\pi_{1,1}, \dots, \pi_{1,n_1})$, and its policy space $\boldsymbol{\Pi}_{1,\text{hete}}$, the meta policy $\sigma_{1,\text{hete}} \in \Delta\boldsymbol{\Pi}_{1,\text{hete}}^r$ under the PSRO framework is a probabilistic strategy over the restricted policy population $\boldsymbol{\Pi}_1^r = \{\vec{\boldsymbol{\pi}}_1^1, \vec{\boldsymbol{\pi}}_1^2, \dots, \vec{\boldsymbol{\pi}}_1^n\}$ with $\vec{\boldsymbol{\pi}}_1^i \in \boldsymbol{\Pi}_{1,\text{hete}}, \forall i \in \{1, \dots, n\}$. Together with the meta policy $\sigma_{2,\text{hete}} \in \Delta\boldsymbol{\Pi}_{2,\text{hete}}^r$, the space of restricted meta policies $\Delta\boldsymbol{\Pi}_{1,\text{hete}} \times \Delta\boldsymbol{\Pi}_{2,\text{hete}}$ can cover equilibrium set **E**, and thus guarantee an global TMECor.

For example, consider a condition where $\pi_{1,i}$ and $\pi_{2,j}$ represent deterministic policies. With the iteration of the PSRO framework going (see details in Section 4.4), the restricted policy set $\boldsymbol{\Pi}_{1,\text{hete}}^r$ and $\boldsymbol{\Pi}_{2,\text{hete}}^r$ expands. When $\boldsymbol{\Pi}_{1,\text{hete}}^r$ and $\boldsymbol{\Pi}_{2,\text{hete}}^r$ grow to contain all deterministic policies, then the space of distributions over the restricted policy sets $\Delta\boldsymbol{\Pi}_{1,\text{hete}}^r$ and $\Delta\boldsymbol{\Pi}_{2,\text{hete}}^r$ can represent any probabilistic policy over the team policy space $\boldsymbol{\Pi}_1$ and $\boldsymbol{\Pi}_2$. This is to say, meta policy $\sigma_{1,\text{hete}}$ and $\sigma_{2,\text{hete}}$ can represent any joint policy of team $T_1$ and opponent team $T_2$. As a result, with iteration going, $\Delta\boldsymbol{\Pi}_{1,\text{hete}}^r \times \Delta\boldsymbol{\Pi}_{2,\text{hete}}^r$ can cover the whole TMECor set **E**, and therefore is able to achieve the global TMECor. $\square$

### A.3 PROOF OF BETTER EX ANTE COORDINATION OF SEQUENTIAL BRO

**Theorem 2** Given an opponent team policy $\boldsymbol{\pi}_2 \in \boldsymbol{\Pi}_2$ (or a team policy $\boldsymbol{\pi}_1 \in \boldsymbol{\Pi}_1$), the sequential BRO can achieve better *ex ante* team coordination than the policy sharing based BRO with $R_1(\mathbf{BR}_{1,\text{seq}}(\boldsymbol{\pi}_2), \boldsymbol{\pi}_2) \geq R_1(\mathbf{BR}_{1,\text{share}}(\boldsymbol{\pi}_2), \boldsymbol{\pi}_2)$ and $R_2(\boldsymbol{\pi}_1, \mathbf{BR}_{2,\text{seq}}(\boldsymbol{\pi}_1)) \geq R_2(\boldsymbol{\pi}_1, \mathbf{BR}_{2,\text{share}}(\boldsymbol{\pi}_1))$, where $\mathbf{BR}_{k,\text{share}} : \boldsymbol{\Pi}_{-k} \to \boldsymbol{\Pi}_{k,\text{share}}$ is the policy sharing based BRO of team $T_k \in \mathcal{T}$, and $\mathbf{BR}_{k,\text{seq}} : \boldsymbol{\Pi}_{-k} \to \boldsymbol{\Pi}_{k,\text{seq}}$ is the sequential BRO of team $T_k \in \mathcal{T}$. In some cases, $R_1(\mathbf{BR}_{1,\text{seq}}(\boldsymbol{\pi}_2), \boldsymbol{\pi}_2) > R_1(\mathbf{BR}_{1,\text{share}}(\boldsymbol{\pi}_2), \boldsymbol{\pi}_2)$ and $R_2(\boldsymbol{\pi}_1, \mathbf{BR}_{2,\text{seq}}(\boldsymbol{\pi}_1)) > R_2(\boldsymbol{\pi}_1, \mathbf{BR}_{2,\text{share}}(\boldsymbol{\pi}_1))$ hold.

*Proof.* Given a meta policy of opponent team $T_2$ (or of team $T_1$) $\boldsymbol{\pi}_2$ (or $\boldsymbol{\pi}_1$), the Best Response computed by sequential BRO is $\mathbf{BR}_{1,\text{seq}}(\boldsymbol{\pi}_2)$ (or $\mathbf{BR}_{2,\text{seq}}(\boldsymbol{\pi}_1)$) and the Best Response computed by sequential BRO is $\mathbf{BR}_{1,\text{share}}(\boldsymbol{\pi}_2)$ (or $\mathbf{BR}_{2,\text{share}}(\boldsymbol{\pi}_1)$). Then $R_1(\mathbf{BR}_{1,\text{seq}}(\boldsymbol{\pi}_2), \boldsymbol{\pi}_2) \geq R_1(\mathbf{BR}_{1,\text{share}}(\boldsymbol{\pi}_2), \boldsymbol{\pi}_2)$, and similarly $R_2(\boldsymbol{\pi}_1, \mathbf{BR}_{2,\text{seq}}(\boldsymbol{\pi}_1)) \geq R_2(\boldsymbol{\pi}_1, \mathbf{BR}_{2,\text{share}}(\boldsymbol{\pi}_1))$. This is because: 1) when the best response $\mathbf{BR}(\boldsymbol{\pi}_2) = \arg\max R_1(\mathbf{BR}(\boldsymbol{\pi}_2), \boldsymbol{\pi}_2) \in \boldsymbol{\Pi}_{1,\text{share}} \subsetneqq \boldsymbol{\Pi}_1$, $\mathbf{BR}(\boldsymbol{\pi}_2) = \mathbf{BR}_{1,\text{seq}}(\boldsymbol{\pi}_2) = \mathbf{BR}_{1,\text{share}}(\boldsymbol{\pi}_2)$ and therefore $R_1(\mathbf{BR}_{1,\text{seq}}(\boldsymbol{\pi}_2), \boldsymbol{\pi}_2) = R_1(\mathbf{BR}_{1,\text{share}}(\boldsymbol{\pi}_2), \boldsymbol{\pi}_2)$; 2) when the best response $\mathbf{BR}(\boldsymbol{\pi}_2) = \arg\max R_1(\mathbf{BR}(\boldsymbol{\pi}_2), \boldsymbol{\pi}_2) \in \boldsymbol{\Pi}_1 \backslash \boldsymbol{\Pi}_{1,\text{share}}$, $\mathbf{BR}_{1,\text{seq}}(\boldsymbol{\pi}_2) \neq \mathbf{BR}_{1,\text{share}}(\boldsymbol{\pi}_2)$ and $R_1(\mathbf{BR}_{1,\text{seq}}(\boldsymbol{\pi}_2), \boldsymbol{\pi}_2) > R_1(\mathbf{BR}_{1,\text{share}}(\boldsymbol{\pi}_2), \boldsymbol{\pi}_2)$. Since the team policy set $\boldsymbol{\Pi}_1 \backslash \boldsymbol{\Pi}_{1,\text{share}} \neq \emptyset$ and the opponent team policy set $\boldsymbol{\Pi}_2 \backslash \boldsymbol{\Pi}_{2,\text{share}} \neq \emptyset$ (Proposition 1), the sequential BRO achieves better *ex ante* team coordination than the BRO with policy sharing with $R_1(\mathbf{BR}_{1,\text{seq}}(\boldsymbol{\pi}_2), \boldsymbol{\pi}_2) > R_1(\mathbf{BR}_{1,\text{share}}(\boldsymbol{\pi}_2), \boldsymbol{\pi}_2)$ and $R_2(\boldsymbol{\pi}_1, \mathbf{BR}_{2,\text{seq}}(\boldsymbol{\pi}_1)) > R_2(\boldsymbol{\pi}_1, \mathbf{BR}_{2,\text{share}}(\boldsymbol{\pi}_1))$ holding in the second case. $\square$

### A.4 PROOF OF CONVERGENCE OF H-PSRO

**Theorem 3** In *heterogeneous* team games, H-PSRO achieves lower exploitability than Team PSRO. Formally, $e(\sigma_{1,\text{seq}}^*, \sigma_{2,\text{seq}}^*) \leq e(\sigma_{1,\text{share}}^*, \sigma_{2,\text{share}}^*)$.

*Proof.* According to definition, $\sigma^*_{\text{share}} = (\sigma^*_{1,\text{share}}, \sigma^*_{2,\text{share}})$ is a meta TMECor within the joint policy space $\mathbf{\Pi}_{1,\text{share}} \times \mathbf{\Pi}_{2,\text{share}}$ (see Proposition 3 in Appendix A.1), and $\sigma^*_{\text{seq}} = (\sigma^*_{1,\text{seq}}, \sigma^*_{2,\text{seq}})$ is a meta TMECor within the joint policy space $\mathbf{\Pi}_1 \times \mathbf{\Pi}_2$ (see Theorem 1 in Appendix A.2). We prove the theorem from two different cases: 1) if $\sigma^*_{\text{share}}$ is a global TMECor within $\mathbf{\Pi}_1 \times \mathbf{\Pi}_2$, then $e(\sigma^*_{1,\text{share}}, \sigma^*_{2,\text{share}}) = e(\sigma^*_{1,\text{seq}}, \sigma^*_{2,\text{seq}})$ since $\sigma^*_{\text{seq}}$ is also a global TMECor; 2) however, $\sigma^*_{\text{share}}$ may not be a global TMECor with $\mathbf{\Pi}_{k,\text{share}} \subsetneqq \mathbf{\Pi}_k$ holding for all $T_k \in \mathcal{T}$ (see Lemma 2 in Appendix A.1). In case that $\sigma^*_{\text{share}}$ is not a global TMECor, $e(\sigma^*_{1,\text{seq}}, \sigma^*_{2,\text{seq}}) = 0$ and $e(\sigma^*_{1,\text{share}}, \sigma^*_{2,\text{share}}) > 0$, making $e(\sigma^*_{1,\text{seq}}, \sigma^*_{2,\text{seq}}) < e(\sigma^*_{1,\text{share}}, \sigma^*_{2,\text{share}})$ hold. As a result, $e(\sigma^*_{1,\text{seq}}, \sigma^*_{2,\text{seq}}) \leq e(\sigma^*_{1,\text{share}}, \sigma^*_{2,\text{share}})$, and $e(\sigma^*_{1,\text{seq}}, \sigma^*_{2,\text{seq}}) < e(\sigma^*_{1,\text{share}}, \sigma^*_{2,\text{share}})$ in the second case.

$\square$

## B    ABLATION STUDIES

We illustrate how the performance evolves for H-PSRO and other baseline methods using the MAgent game (Zheng et al., 2018; Terry et al., 2020) in Appendix B.1, where H-PSRO is more effective at approximating a TMECor with the enlarging task scales. An ablation study on relative performance against state-of-the-art MARL algorithms of H-PSRO in Appendix B.2 reveals that, with different MARL opponent strategies, H-PSRO exhibits superior win rate and more steady performance. The competitive videos are available at `https://sites.google.com/view/h-psro-2024/h-psro`.

### B.1    HOMOGENEOUS TEAM GAME

MAgent Battle (Zheng et al., 2018; Terry et al., 2020) is a gridworld game where a red team of $N$ homogeneous agents fight against a blue homogeneous team. At each step, agents can move to one of the 12 nearest grids or attack one of the 8 surrounding grids of themselves. The game terminates if all agents in the same team are killed or reaches a maximum number of steps. To compare the scalability of H-PSRO, Team PSRO (McAleer et al., 2023) and PSRO (Lanctot et al., 2017) in homogeneous team games, we run algorithms in MAgent Battle games of different scales, including 6-vs-6, 12-vs-12, 16-vs-16. Since the exploitability cannot be exactly calculated in this games, we estimate the Single Side Reward (SSR) of the final equilibrium policies against random policies and differently correlated Best Response as opponent team policies. The averaged results over 3 seeds are shown in Table 1.

Notably, H-PSRO agents achieve the lowest SSR in large scale MAgent Battles (e.g., 12-vs-12 and 16-vs-16) and comparable performance to PSRO and Team PSRO in mediated scale games (e.g., 6-vs-6). This is because in mediated scale homogeneous team games, such as 6-vs-6 MAgent Battle, TMECor can be found by enumerating all possible attacking strategies with PSRO. However, in larger scale games, the policy space (see Table 1) becomes exponentially enormous, making PSRO methods very inefficient. On the other hand, the impacts of insufficient policy expressive ability (see Proposition 1 in Appendix A.1) becomes more severe as the game scale increases, making Team PSRO, though efficient, struggle to approximate the global TMECor in large homogeneous team games.

### B.2    HETEROGENEOUS PSRO VS MARL ALGORITHMS

We compare the win rate of H-PSRO and Team PSRO (McAleer et al., 2023) against several state-of-the-art MARL algorithms, including MAPPO (Yu et al., 2022), HAPPO (Kuba et al., 2022), and MAT (Wen et al., 2022) in Competitive StarCraft Benchmark (Leroy et al., 2022). The experimental results are shown in Table 2, where H-PSRO achieves significantly higher win rate than Team PSRO when they are against HAPPO and MAT, and achieves comparable win rate of approximate 100 when they are against the homogeneous coordination algorithm MAPPO, which inherits an insufficient policy expressive ability (see Proposition 1 in Appendix A.1). We also observe that H-PSRO achieves relative steady performance against diverse opponent strategies while the MARL algorithms and Team PSRO suffer from severe performance instability, indicating a sub-optimal TMECor.

Table 1: Performance of H-PSRO, Team-PSRO and PSRO in MAgent (Zheng et al., 2018; Terry et al., 2020). MAgent (Zheng et al., 2018; Terry et al., 2020) is a gridworld battle scenario where each player has 21 actions. When increasing the number of teammates, the team joint action space explodes exponentially. We show in larger games (e.g., 12v12, 16v16), H-PSRO is capable of finding equilibrium policies with lower SSR when confronting opponent teams with different exploitation ability. Due to the symmetric team setting, we use a metric named Single Side Reward (SSR) $SSR(\pi_1, \pi_2) = 2R_1(\pi_1, \mathrm{BR}(\pi_1))$ to measure the performance of the population.

| GAME SETTING | TEAM JOINT ACTION SPACE | ALGORITHM | SSR (WINRATE) OVER DIFFERENT OPPONENTS | | | | |
|---|---|---|---|---|---|---|---|
| | | | SEQUENTIAL CORRELATION | JOINT CORRELATION | SYNCHRONIZED CORRELATION | NO CORRELATION | RANDOM |
| 6v6 | 8.58E+7 | H-PSRO | 20.223 (0.66) | 11.074 (0.48) | 11.153 (0.56) | 7.01 (0.43) | -4.640 (0) |
| | | TEAM PSRO | 23.877 (0.73) | 18.390 (0.62) | 13.581(0.56) | 14.842 (0.61) | -2.980 (0) |
| | | **PSRO** | **13.439 (0.56)** | **6.691 (0.33)** | **3.263 (0.11)** | **6.302 (0.26)** | **-5.377 (0)** |
| 12v12 | 7.36E+15 | H-PSRO | **12.964 (0.55)** | **-1.172 (0)** | **-2.062 (0.01)** | **0.403 (0.14)** | **-7.749 (0)** |
| | | TEAM PSRO | 28.182 (0.69) | 4.931 (0.24) | 6.676 (0.32) | 16.060 (0.55) | -4.650 (0.01) |
| | | PSRO | 16.222 (0.55) | 2.488 (0.08) | 2.138 (0.24) | 7.418 (0.33) | -4.992 (0) |
| 16v16 | 1.43E+21 | H-PSRO | **13.449 (0.43)** | **-1.711 (0.03)** | **-10.198 (0.09)** | **-0.563 (0.24)** | **-6.854 (0.01)** |
| | | TEAM PSRO | 25.412 (0.60) | -1.454 (0.01) | -7.941 (0.13) | 16.767 (0.48) | -3.394 (0.01) |
| | | PSRO | 26.929 (0.80) | 0.597 (0.04) | 6.396 (0.37) | 22.239 (0.69) | -2.656 (0) |

Table 2: Performance of H-PSRO, Team-PSRO in Competitive StarCraft2 (Leroy et al., 2022). Competitive StarCraft2 is a battle game where each team consists of units from three species (Marines, Stalkers, Zealots), and each unit has 9 actions. We consider heterogeneous scenarios where the units within each team are heterogeneous and the units in both teams are the symmetric, as shown below. We evaluate H-PSRO and Team PSRO by comparing the win rate against the strategies of several state-of-the-art MARL algorithms, including MAT (Wen et al., 2022), HAPPO (Kuba et al., 2022), and MAPPO (Yu et al., 2022).

| MAPS | TYPE | TEAM UNITS | ALGORITHM | WIN RATE OVER DIFFERENT OPPONENTS | | |
|---|---|---|---|---|---|---|
| | | | | MAT | HAPPO | MAPPO |
| 2S3Z_COMPETE (5V5) | HETEROGENEOUS & SYMMETRIC | 2 STALKERS & 3 ZEALOTS | H-PSRO | **34.0** | **56.0** | 98.0 |
| | | | TEAM PSRO | 7.0 | 7.0 | **100.0** |
| 3S5Z_COMPETE (8V8) | HETEROGENEOUS & SYMMETRIC | 3 STALKERS & 5 ZEALOTS | H-PSRO | **89.0** | **72.0** | 99.0 |
| | | | TEAM PSRO | 18.0 | 1.0 | **100.0** |
| MMM_COMPETE (10V10) | HETEROGENEOUS & SYMMETRIC | 1 MEDIVAC, 2 MARAUDERS & 7 MARINES | H-PSRO | **59.0** | **85.0** | **100.0** |
| | | | TEAM PSRO | 20.0 | 10.0 | 95.0 |

## C  EXAMPLES OF CONVERGENCE ISSUE IN HETEROGENEOUS TEAM GAMES

**Example 2.** Consider a heterogeneous team game with two teams $T_1 = \{M_1, M_2\}$, $T_2 = \{O_1, O_2\}$, one state and joint action spaces $\mathcal{A}_1 = \{0, 1\} \times \{0, 2\}$, $\mathcal{A}_2 = \{0, 1\} \times \{0, 3\}$, where the reward is given by:

$$R_1 = \begin{cases} 4 & \pi_{1,M_1}^{(0)} = \pi_{1,M_2}^{(2)} = 1, \pi_{2,O_1}^{(0)} = \pi_{2,O_2}^{(0)} = 1, \\ \nu_2 - \nu_1 + 1 & otherwise, \end{cases} \tag{5a}$$

$$R_2 = -R_1. \tag{5b}$$

where $\nu_1 = 2\pi_{1,M_1}^{(1)} + 2\pi_{1,M_2}^{(2)}$ and $\nu_2 = 2\pi_{2,O_1}^{(1)} + 3\pi_{2,O_2}^{(3)}$. Here, $\pi_{1,M_1}^{(0)}$ denotes the probability of action 0 for player $M_1 \in T_1$. An TMECor in this case is a probabilistic policy over both teams' joint action space: team $T_1$ takes joint action $(0,0)$ with probability 0.6, $(0,2)$ with probability 0.4, and other actions with probability 0, and opponent team $T_2$ takes $(0,0)$ with probability 0.4, $(1,0)$ with probability 0.6 and other actions with probability 0. However, we show that policy sharing among teammates constrains the team joint policies to a small subset of the entire policy space, and excludes the above TMECor solution. A shared policy is a vector of shared action distribution, which can be denoted by $\pi_{1,\text{share}} = (x_1, x_2)$ or $\pi_{2,\text{share}} = (y_1, y_2)$. With a shared action distribution, the team joint policy will be constrained to a subset of the whole joint action distribution denoted by $\boldsymbol{\Pi}_{1,\text{share}} = \{(x_1^2, x_1 x_2, x_2 x_1, x_2^2)|x_1 \in [0,1], x_2 \in [0,1], x_1 + x_2 = 1.0\} \subsetneq \Delta\mathcal{A}_1$ or $\boldsymbol{\Pi}_{2,\text{share}} = \{(y_1^2, y_1 y_2, y_2 y_1, y_2^2)|y_1 \in [0,1], y_2 \in [0,1], y_1 + y_2 = 1.0\} \subsetneq \Delta\mathcal{A}_2$, in which the probability of joint action $(0,2)$ is constrained to be *equal* to the probability of joint action $(1,0)$ for team $T_1$. However, this conflicts with the TMECor strategy of team $T_1$, where the probability of joint action $(0,2)$ is 0.4, and the probability of joint action $(1,0)$ is 0.0.

# D ANALYSIS OF EXISTING WORK

Team games are addressed from three different perspectives: the competitive perspective, the cooperative perspective, and the mixed cooperative-competitive perspective.

## D.1 COMPETITIVE PERSPECTIVE

From the competitive perspective, an entire team is treated as a single player with a joint action space, effectively transforming a two-team zero-sum game into a two-player zero-sum game (Farina et al., 2018; Carminati et al., 2022). Consequently, finding a TMECor in a two-team zero-sum game is equivalent to finding a Nash equilibrium in a two-player zero-sum game.

**PSRO.** Policy Space Response Oracles (PSRO) (Lanctot et al., 2017; McAleer et al., 2020; Liu et al., 2021) have been widely used to approximate Nash equilibria in large-scale two-player zero-sum games and can be adapted to equivalent team games to approximate TMECor. To manage the large policy space, PSRO incrementally develops a population of joint team policies to approximate the whole team joint policy space (e.g., $\boldsymbol{\Pi}_1, \boldsymbol{\Pi}_2$). Initially, PSRO begins with a population $\boldsymbol{\Pi}_k^r = \{\boldsymbol{\pi}_k^1\}$ for team $T_k \in \mathcal{T}$, which consists of a single randomly generated joint policy parameterized by $\vartheta_k$. At each iteration $t$, an empirical payoff matrix $U$ is derived from simulations of current population $\boldsymbol{\Pi}_k^r$ and $\boldsymbol{\Pi}_{-k}^r$. This payoff matrix $U$ is then utilized by a meta-solver to determine the meta-policy $\sigma_k$, and a new policy $\boldsymbol{\pi}_{-k}$, parameterized by $\vartheta_{-k}$ is trained to be the best response (BR) to the meta policy $\sigma_k$. Then new policy $\boldsymbol{\pi}_{-k}$ is added to the population $\boldsymbol{\Pi}_k^r$, and the process repeats. When the newly trained BR already exists in the population, PSRO outputs a final distribution over the population policies, effectively approximating the TMECor of the original team game.

A significant challenge from the competitive perspective is that transforming a two-team zero-sum game into a two-player zero-sum game causes *the equilibrium search space to grow exponentially with the number of players in both teams*. This makes directly applying PSRO to solve TMECor infeasible for large team games.

## D.2 COOPERATIVE PERSPECTIVE

Another perspective for solving TMECor is to model two team games as cooperative games and treat the opposing team $T_2$ part of the environment. From this viewpoint, solving TMECor equates to maximizing the following objective:

$$J(\boldsymbol{\pi}_1) \triangleq R_1(\boldsymbol{\pi}_1, \cdot).$$

When the objective achieves its maximal value, no other strategy $\boldsymbol{\pi}_1 \in \boldsymbol{\Pi}_1$ can yield a higher reward, indicating that team $T_1$ has reached a TMECor. In the cooperative perspective, the challenge lies in how to coordinate teammates within $T_1$ while ensuring convergence to TMECor. To solve this, various Multi-Agent Reinforcement Learning (MARL) algorithms (De Witt et al., 2020; Yu et al., 2022; Kuba et al., 2022; Wen et al., 2022) have been proposed. Within these approaches, players in $T_1$ take the actions with the maximal value of the state-action value function $Q_{\boldsymbol{\pi}_1}(\boldsymbol{o}_1, \boldsymbol{a}_1)$, which is defined as:

$$Q_{\boldsymbol{\pi}_1}(\boldsymbol{o}_1, \boldsymbol{a}_1) \triangleq \mathbb{E}_{\mathbf{o}_{1,1:\infty} \sim P, \mathbf{a}_{1,1:\infty} \sim \boldsymbol{\pi}_1} \left[ R_1^\gamma \mid \mathbf{o}_{1,0} = \boldsymbol{o}_1, \mathbf{a}_{1,0} = \boldsymbol{a}_0 \right].$$

The advantage function of $\boldsymbol{\pi}_1$ is defined to be

$$A_{\boldsymbol{\pi}_1}(\boldsymbol{o}_1, \boldsymbol{a}_1) \triangleq Q_{\boldsymbol{\pi}_1}(\boldsymbol{o}_1, \boldsymbol{a}_1) - V_{\boldsymbol{\pi}_1}(\boldsymbol{o}_1), \tag{6}$$

and $V_{\boldsymbol{\pi}_1}(\boldsymbol{o}_1)$ is the observation value function defined as[3]:

$$V_{\boldsymbol{\pi}_1}(\boldsymbol{o}_1) \triangleq \mathbb{E}_{\mathbf{a}_{1,0:\infty} \sim \boldsymbol{\pi}_1, \mathbf{o}_{1,1:\infty} \sim P} \left[ R_1^\gamma \mid \mathbf{o}_{1,0} = \boldsymbol{o}_1 \right]$$

**MAPPO.** MAPPO (Yu et al., 2022) coordinates players in $T_1$ by extending PPO (Schulman et al., 2017) to multiple players. To do this, MAPPO employs a trick of policy sharing, where all agents in team $T_1$ share a policy $\pi_{1,\text{share}}$, so that $\vec{\boldsymbol{\pi}}_{1,\text{share}} = (\pi_{1,\text{share}}, \dots, \pi_{1,\text{share}})$ (De Witt et al., 2020; Yu et al., 2022). As such, the policy is updated to maximise

$$\mathcal{L}^{\text{MAPPO}}(\pi_{1,\text{share}}) \triangleq \mathbb{E}_{\boldsymbol{o}_1 \sim \rho_{\pi_{\text{old}}}, \boldsymbol{a}_1 \sim \pi_{\text{old}}} \left[ \sum_{i=1}^{n_1} \min \left( \frac{\pi(a_1^i | o_1^i)}{\pi_{\text{old}}(a_1^i | o_1^i)} A_{\pi_{\text{old}}}(\boldsymbol{o}_1, \boldsymbol{a}_1), \text{clip} \left( \frac{\pi(a_1^i | o_1^i)}{\pi_{\text{old}}(a_1^i | o_1^i)}, 1 \pm \epsilon \right) A_{\pi_{\text{old}}}(\boldsymbol{o}_1, \boldsymbol{a}_1) \right) \right], \tag{7}$$

---

[3]We write $\boldsymbol{a}_{1,t}^i$, $\boldsymbol{a}_{1,t}$ and $\boldsymbol{o}_{1,t}$ when we refer to the action, joint action and joint observation as to values, and $\mathbf{a}_{1,t}^i$, $\mathbf{a}_{1,t}$ and $\mathbf{o}_{1,t}$ as to random variables.

where the clip$(, 1 \pm \epsilon)$ operator clips the input to $1 - \epsilon/1 + \epsilon$ if it is below/above this value, thereby preventing large policy updates and stabling the training process. Indeed, the algorithm does not introduce much computational burden with the increasing number of teammates $|T_1|$. Nevertheless, the policy-sharing team strategy limits the algorithm's applicability and could lead to its suboptimality (Kuba et al., 2022; Zhong et al., 2024) when agents have different roles.

**HAPPO.** To handle this, Heterogeneous Agent Proximal Policy Optimization (HAPPO) (Kuba et al., 2022) was proposed. Instead of coordinating agents by sharing one policy among them, HAPPO parameterizes each agent's policy $\pi_{1,\vartheta_i}(\pi_{1,i})$ by $\vartheta_i$, which, together with other agents' policies, forms a joint team policy $\vec{\pi}_{1,\vartheta_1}(\vec{\pi}_1)$ parameterized by $\vartheta_1 = (\vartheta_1, \dots, \vartheta_{n_1})$. To optimise the $\vartheta_1$, HAPPO follows the idea of PPO by considering only using first-order derivatives. This is achieved by making agent $i_m \in T_1$ choose a policy parameter $\vartheta_{i_m}^{k+1}$ which maximises the clipping objective of

$$
\mathbb{E}_{\mathbf{o}_1 \sim \rho_{\vec{\pi}_{1,\vartheta_1^k}}, \mathbf{a}_1^{i_{1:m-1}} \sim \vec{\pi}_{1,\vartheta_{i_{1:m-1}}^{k+1}}, \mathbf{a}_1^{i_m} \sim \pi_{1,\vartheta_{i_m}^k}} \left[ \min \left( \mathrm{r}(\pi_{1,\vartheta_{i_m}^{k+1}}) A_{\vec{\pi}_{1,\vartheta_1^k}}^{i_{1:m}}(\boldsymbol{o}_1, \boldsymbol{a}_1^{i_{1:m}}), \mathrm{clip}(\mathrm{r}(\pi_{1,\vartheta_{i_m}^{k+1}}), 1 \pm \epsilon) A_{\vec{\pi}_{1,\vartheta_1^k}}^{i_{1:m}}(\boldsymbol{o}_1, \boldsymbol{a}_1^{i_{1:m}}) \right) \right],
$$
(8)

where $\mathrm{r}(\pi_{1,\vartheta_m^{k+1}}) = \pi_{1,\vartheta_m^{k+1}}(\mathrm{a}_1^{i_m} \mid \mathbf{o}_1)/\pi_{1,\vartheta_m^k}(\mathrm{a}_1^{i_m} \mid \mathbf{o}_1)$ and $A_{\vec{\pi}_{1,\vartheta_1^k}}^{i_{1:m}}(\boldsymbol{o}_1, \boldsymbol{a}_1^{i_{1:m}})$ is the multi-agent advantage function (Kuba et al., 2022; Wang et al., 2023) defined as:

$$
A_{\vec{\pi}_{1,\vartheta_1^k}}^{i_{1:m}}(\boldsymbol{o}_1, \boldsymbol{a}_1^{i_{1:m}}) \triangleq \sum_{j=1}^{m} A_{\vec{\pi}_{1,\vartheta_1^k}}^{i_j}\left(\boldsymbol{o}_1, \boldsymbol{a}_1^{i_{1:j-1}}, a_1^{i_j}\right).
$$
(9)

Based on the Multi-Agent Advantage Decomposition Theorem (Kuba et al., 2022), HAPPO is proven to enjoy monotonic improvement and guaranteed convergence to the Nash Equilibrium (NE) when environmental conditions keep stable and opponent team strategies stay invariant.

**MAT.** Following this, Wen et al. (2022) take effort to build a connection between multi-agent reinforcement learning (MARL) problems and generic sequence models (SM), and propose Multi-Agent Transformer (MAT), which leverages transformer architectures to model complex interactions between cooperative players in team $T_1$.

While the aforementioned algorithms have demonstrated remarkable performance in team games such as StarCraft II (Samvelyan et al., 2019), this performance is achieved when the opponent team is fixed. Extending these algorithms to broader scenarios, such as when encountering different human opponent teams, remains a significant challenge.

### D.3 MIXED COOPERATIVE-COMPETITIVE PERSPECTIVE

To address the challenges from both competitive and cooperative perspectives and to approximate TMECor in large-scale team games without losing generality, McAleer et al. (2023) extend the PSRO framework by integrating a homogeneous cooperative reinforcement learning techniques (e.g., MAPPO), and propose a homogeneous PSRO framework named Team PSRO. Specifically, it iteratively constructs a population of shared policies $\Pi_{k,\mathrm{share}}^r = \{\vec{\pi}_{k,\mathrm{share}}^1, \dots, \vec{\pi}_{k,\mathrm{share}}^n\}$, where $\vec{\pi}_{k,\mathrm{share}}^i = (\pi_{k,\mathrm{share}}^i, \dots, \pi_{k,\mathrm{share}}^i) \in \Pi_{k,\mathrm{share}}$, by adding the best response to the meta-policy over $\Pi_{k,\mathrm{share}}^r$ via Eq (7). Team PSRO eventually converges to a TMECor within $\Pi_{1,\mathrm{share}} \times \Pi_{2,\mathrm{share}}$, maintaining robustness against various opponent teams while not imposing additional computational burden as the number of players in both teams increases. However, as analyzed in Section 4.1 and Section 4.2, this homogeneous framework may encounter convergence issues in heterogeneous team games, including terminating early and never converges to the global TMECor, and being trapped into a sub-optimal point.

## E ALGORITHM

---

**Algorithm 2:** SequentialBRO

---

1 **input :** Unrestricted Policy Space $\mathbf{\Pi}_{k,\mathrm{seq}}$, Restricted Policy Space $\mathbf{\Pi}^r_{-k,\mathrm{hete}}$, Prefixed meta strategy of opposing team $\sigma_{-k,\mathrm{seq}} \sim \mathbf{\Pi}^r_{-k,\mathrm{hete}}$

2 **input :** Stepsize $\alpha$, batch size $B$, number of: agents $n$, episodes $P$, steps per episode $T$.

3 **Initialize :** Actor networks $\boldsymbol{\vartheta} = \{\vartheta_i, \forall i \in T_k, T_k \in \mathcal{T}\}$, optimal V-value network $\{\phi_0\}$, Replay buffer $\mathcal{B}$

4 **for** $p = 0, 1, \ldots, P-1$ **do**

5 $\quad$ Collect a set of trajectories by running the team policy $\vec{\boldsymbol{\pi}}_{k,\boldsymbol{\vartheta}} = \left(\pi_{k,\vartheta_1}, \ldots, \pi_{k,\vartheta_{n_k}}\right)$ and prefixed opposing team policy $\sigma_{-k,\mathrm{seq}}$.

6 $\quad$ Push transitions $\left\{\left(o^i_{k,t}, a^i_{k,t}, o^i_{k,t+1}, r_{k,t}\right), \forall i \in T_k, t \in T\right\}$ into $\mathcal{B}$.

7 $\quad$ Sample a random minibatch of $B$ transitions from $\mathcal{B}$.

8 $\quad$ Compute advantage function $\hat{A}(\boldsymbol{o}_k, \mathbf{a}_k)$ based on optimal V-value network with GAE.

9 $\quad$ Draw a random permutation of agents $i_{1:n_k}$.

10 $\quad$ Set $M^i(\boldsymbol{o}_k, \mathbf{a}_k) = \hat{A}(\boldsymbol{o}_k, \mathbf{a}_k)$.

11 $\quad$ **for** $i \in T_k$ **do**

12 $\quad\quad$ Update policy parameter $\vartheta^{p+1}_i$ with argmax of the objective

$$\arg\max_{\vartheta^p_i} \frac{1}{BT} \sum_{b=1}^{B} \sum_{t=0}^{T} \min\left(\frac{\pi_{k,\vartheta_i}\left(a^i_{k,t}|o^i_{k,t}\right)}{\pi_{k,\vartheta^p_i}\left(a^i_{k,t}|o^i_{k,t}\right)} M^{1:i}(\boldsymbol{o}_k, \mathbf{a}_k), \mathrm{clip}\left(\frac{\pi_{k,\vartheta_i}\left(a^i_{k,t}|o^i_{k,t}\right)}{\pi_{k,\vartheta^p_i}\left(a^i_{k,t}|o^i_{k,t}\right)}, 1\pm\epsilon\right) M^{1:i}(\boldsymbol{o}_k, \mathbf{a}_k)\right).$$

13 $\quad\quad$ Compute $M^{1:i+1}(\boldsymbol{o}_k, \mathbf{a}_k) = \frac{\pi_{k,\vartheta^{p+1}_i}\left(\mathrm{a}^i_k|o^i_k\right)}{\pi_{k,\vartheta^p_i}\left(\mathrm{a}^i_k|o^i_k\right)} M^{1:i}(\boldsymbol{o}_k, \mathbf{a}_k)$.

14 $\quad\quad$ Update $\pi_{k,\vartheta_i} \leftarrow \pi_{k,\vartheta^{p+1}_i}$.

15 $\quad$ Update V-value network by following formula:

$$\phi_{p+1} = \arg\min_{\phi} \frac{1}{BT} \sum_{b=1}^{B} \sum_{t=0}^{T} \left(V_\phi\left(\boldsymbol{o}_k\right) - \hat{R}_t\right)^2$$

16 **output :** $T_k$'s sequentially correlated best response strategy $\vec{\boldsymbol{\pi}}_{k,\boldsymbol{\vartheta}}$

---

