# OpenReview forum: "Computing Ex Ante Equilibrium in Heterogeneous Zero-Sum Team Games"
_ICLR.cc/2025/Conference — ICLR 2025 Conference Withdrawn Submission_

### Official Review · Reviewer_bSXu · 2024-10-24

**Soundness:** 1
**Presentation:** 1
**Contribution:** 1
**Rating:** 3
**Confidence:** 5

**Summary:**

This paper proposes an algorithm H-PSRO to compute TMECor Equilibrium in heterogeneous zero-sum team games by parameterizing each player's policy. The design of H-PSRO algorithm is based on previous work Team PSRO. This work also provides some theoretical analysis of the proposed algorithm along with some numerical results to demonstrate its practical performance.

**Strengths:**

1. Numerical results are provided where the proposed algorithm is competing with several benchmark algorithms in relatively large-scale two team zero-sum games.

**Weaknesses:**

1. This paper makes the claim that previous work [1] relies on the assumption that the two-team zero-sum has to be homogenous and a policy sharing mechanism that requires every player in the same team to play under the same strategy. (i.e. the policies of players in the same team are ($\pi_{k, share}, \pi_{k, share}, ..., \pi_{k, share} )$. However, to the reviewer's best knowledge, the algorithm provided in [1] does not require such strong assumption and does not employ a policy sharing mechanism. Team PSRO algorithm in [1] directly optimize the team strategy over the team's joint strategy space, it outputs a joint policy of each team with potential correlation instead of outputting each players' strategy individually (in this case no correlation would be possible). I would appreciate it if the author specify where the policy sharing mechanism is posted in [1].

2. The definition of set $E_c$ in definition 2 remains ambiguous. When considering possible deviations from an equilibrium in $E_c$, do we only consider possible deviations in $S_c$ or we consider all possible deviations in $S$? Consequently, the proof of proposition 2 regard $E_{share} \subseteq E$ is also unclear.

3. It is unclear how the authors reach the conclusion that $S_{share} \subsetneqq S$. Further, in equation (3a) and (3b), the notation $\sigma_{1, share} \in \Pi_1 \backslash \Pi_{1, share}$ remains confusing.

4. Lemma 1 defines the advantage function $A_{\pi_1}^{i_l}(o_1, a_1^{i_{1:l-1}}, a_1^{i_{l}}| \pi_2) = Q_{\pi_1}^{i_{1:l}}(o_1, a_1^{i_{1:l}}| \pi_2) - Q_{\pi_1}^{i_{1:l-1}}(o_1, a_1^{i_{1:l-1}}, a_1^{i_{l}}| \pi_2)$. This notation remains extremely confusing and it's unclear what the value $Q_{\pi_1}^{i_{1:l}}(o_1, a_1^{i_{1:l}}| \pi_2)$ stands for.  The Q-value function in multiagent setting should be evaluated based on the reward of a joint action $a^{i_1:n}$ instead of the joint (or single) action of a subset of players.

5. The authors claim that utilizing the advantage function would bypass the computational difficulty induced by the exponential growth of the the policy space. However, it is unclear how they update the advantage function to bypass this difficulty since the number of advantage function is also growing exponentially with the number of actions.

6. The authors state that the proposed oracle sequential BRO, could efficiently compute the best responding joint policy of a team. However, due to the unclear definition of the advantage function, it's not clear if the oracle can have some monotonic value functions. Furthermore, even some monotonic increment over the value function was achieved by sequentially picking each player's action. It is not sufficient to show that oracle would output the optimal policy.


[1] Stephen Marcus McAleer, Gabriele Farina, Gaoyue Zhou, Mingzhi Wang, Yaodong Yang, and Tuo- mas Sandholm. Team-PSRO for learning approximate tmecor in large team games via cooperative reinforcement learning. In Thirty-seventh Conference on Neural Information Processing Systems, 2023.

**Questions:**

See weaknesses.

---

### Official Review · Reviewer_5S6Q · 2024-11-01

**Soundness:** 2
**Presentation:** 3
**Contribution:** 1
**Rating:** 3
**Confidence:** 5

**Summary:**

The paper claims to have an algorithm that improves on Team-PSRO for computing TMECor in large adversarial team games, by allowing heterogeneous policies between players. They show that theoretically, homogeneous policies are insufficient to capture optimal play; and in experiments, allowing heterogeneous policies improves performance of PSRO.

**Strengths:**

The paper is written clearly. I fully agree with the authors that restricting to homogeneous policies would be severely limiting in the setting of team games.

**Weaknesses:**

This paper is built on the premise that Team-PSRO, as implemented by McAleer et al. (2023), cannot recover heterogeneous policies because it uses shared parameters across team members. This premise is shaky at best, if not outright false. It is very easy to use shared parameters and still allow heterogeneous policies, by using "agent indicators", i.e., including the index of each agent as part of its observation [1, 2]. Although McAleer et al. (2023) does not (as far as I can tell) explicitly state whether or not they are using agent indicators, it is reasonable to guess this from their presentation: throughout the paper, they present the players' policies $\pi_\textsf{T1}, \pi_\textsf{T2}$ as distinct (even belonging to different policy spaces $\Pi_\textsf{T1}, \Pi_\textsf{T2}$), so at least the tabular version of their algorithm certainly allows heterogeneous policies. Moreover, even if McAleer et al (2023) for some reason did not use agent indicators, it is a well-known technique (e.g., [1] has >1k citations), so probably does not deserve an entire paper to point out.

As this is the entire premise of the paper, this is a significant problem and the main reason for my score.

Another comment: the authors claim that "optimizing over multiple players’ policy spaces simultaneously is significantly harder than optimizing over a single shared player’s policy space". This misses the point. In the present paper, the pertinent comparison is not finding an optimal joint policy vs finding an optimal single-player policy. The comparison is between finding an optimal *heterogeneous* joint policy and finding an optimal *homogeneous* joint policy in a *team game*. And here, in fact, the *latter* problem is usually harder. For example, consider two-player identical-interest normal-form games with payoff matrix $M \in \mathbb R^{n \times n}$. Finding an optimal homogeneous joint policy reduces to the quadratic optimization problem $\max_{x \in \Delta_n} x^\top Mx$, which is NP-hard (even to approximate) in general (see e.g. [3]), whereas finding an optimal heterogeneous policy simply requires finding the maximum entry of $M$.

[1] Jayesh K. Gupta, Maxim Egorov, Mykel Kochenderfer (*AAMAS* 2017) "Cooperative Multi-Agent Control Using Deep
Reinforcement Learning"

[2] J K Terry, Nathaniel Grammel, Ananth Hari, Luis Santos, Benjamin Black (2020) "Revisiting Parameter Sharing in Multi-Agent Deep Reinforcement Learning"

[3] Etienne de Klerk (*EJOR* 2008) "The complexity of optimizing over a simplex, hypercube or sphere: a short survey"

**Questions:**

It is clear what a heterogeneous _policy_ is, but what is a "heterogeneous game"? As far as I can tell, this term is never defined, and it seems not very relevant to the paper: for example, for reasonable definitions of "heterogeneous/homogeneous game" that I can think of, e.g., what [4] calls "ordinary player symmetric", it is easy to construct homogeneous games that require heterogeneous policies to play optimally.


[4] Zhigang Cao, Xiaoguang Yang (*Mathematical Social Sciences* 2015) "Symmetric Games Revisited"

---

### Official Review · Reviewer_7n6a · 2024-11-05

**Soundness:** 3
**Presentation:** 3
**Contribution:** 2
**Rating:** 6
**Confidence:** 2

**Summary:**

This paper considers the ex ante equilibrium computation problem in zero-sum heterogeneous team game. The paper first identifies the issues of adopting the existing solver, Policy Space Response Oracle (PSRO), from homogeneous to heterogeneous game. Then, it designs the Heterogeneous-PSRO framework by integrating the sequential correlation mechanism into an iterative procedure.

**Strengths:**

- The paper is well-written, as it clearly explains the problem setup, notions and solutions to me.
- The paper also includes a rich set of experiments on realistic team game instances. The results from the proposed method look good and the plots in the experiments are clear and informative.

**Weaknesses:**

- The insights and technical contributions are relatively shallow. The proposed methods seem to be well-expected, so I would encourage the authors to highlight the challenges in devising and employing the proposed methods.

**Questions:**

Is there any thought on how we can go beyond zero-sum team game, to potential game or monotone game?

---

### Note · Authors · 2024-11-13

I have read and agree with the venue's withdrawal policy on behalf of myself and my co-authors.